# How Graph Neural Networks Learn: Lessons from Training Dynamics in Function Space

## Abstract

A long-standing goal in deep learning has been to characterize the learning behavior of black-box models in a more interpretable manner. For graph neural networks (GNNs), considerable advances have been made in formalizing what functions they can represent, however it remains less clear whether and how GNNs learn desired functions during the optimization process. To fill this critical gap, we study the learning dynamics of GNNs in function space via the analytic framework of overparameterization. In particular, we find that the seemingly complicated training process of GNNs can be re-cast into a more familiar label propagation framework, due to the graph inductive bias implicit in this process. From this vantage point, we provide explanations for why the learned GNN functions successfully generalize and for their pathological behavior on heterophilic graphs, which are consistent with observations. Practically, sparsifying and implementing the learning dynamics lead to a minimalist semi-supervised learning algorithm with the efficiency of classic algorithms and the effectiveness of modern GNNs.

## 1 Introduction

*Graph Neural Networks (GNNs)* (Gori et al., 2005; Scarselli et al., 2008; Bruna et al., 2014; Kipf & Welling, 2017) represent network architectures for learning on entities with explicit relations. In addition to their empirical success, the recent pursuit of theoretical understanding has also led researchers to dissect GNN models, especially regarding their representation (Maron et al., 2019; Xu et al., 2019; Oono & Suzuki, 2019; Chen et al., 2019) and generalization (Scarselli et al., 2018; Verma & Zhang, 2019; Garg et al., 2020) powers. Another research area in parallel with representation and generalization is optimization, which focuses on understanding how the training algorithm identifies desirable GNN functions. Despite its importance, the optimization process of training GNNs is far less understood compared with other topics. While previous work has studied the learning dynamics of linearized GNNs (Xu et al., 2021a) in their *weight space*, with emphasis on the convergence speed of empirical risk, an analysis of their dynamics in *function space* that directly examines how the GNN function evolves has been lacking. In particular, we focus on the following open questions:

①️ *How do GNN models evolve in function space during optimization, and how exactly do graphs regularize or bias this process?*

②️ *In light of the above, are there interpretable ways to explain why the learned GNN function successfully generalizes and when they might fail?*

Studying these questions can lead to deeper understanding of GNNs and provide guidelines for designing more principled algorithms (as we will exemplify in Sec. 3). As an initial probe in this direction, we study GNN learning dynamics in function space, i.e., how the learned function $f_t(\boldsymbol{x})$ (or model prediction) evolves on an arbitrary sample $\boldsymbol{x}$ as model weights $\mathbf{W}_t$ evolves over time according to gradient descent. Our contributions are summarized as follows:

**Theoretical Contributions.** We provide answers to the above two questions, via the widely adopted analytical framework (Jacot et al., 2018; Arora et al., 2019; Du et al., 2019c) of studying neural networks in overparameterized regimes, and map our insights to practice via empirical verification.

○ ***GNN Dynamics Resemble Label Propagation:*** The evolution of GNNs in function space follows a cross-instance propagation scheme with pairwise weights controlled by a kernel matrix $\boldsymbol{\Theta}_t$ called the *Neural Tangent Kernel (NTK)*, which we demonstrate tends to align well with, or even become equivalent to, various specialized graph adjacency matrices $\mathbf{A}$ in overparameterized regimes. In this way,

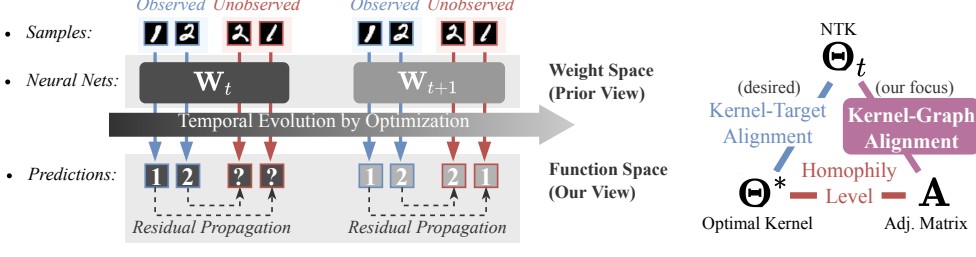

(a) Learning dynamics        (b) Alignment of matrices

Figure 1: **(a)** Learning dynamics of (graph) neural networks from a function space viewpoint where residuals (i.e. difference between ground-truth labels and predictions) propagate from observed to unobserved samples based on a kernel matrix and shape the learned function. (Each image is a data point) **(b)** Alignment of matrices that control the residual propagation process: NTK $\Theta_t$, adjacency matrix $\mathbf{A}$, and the optimal kernel matrix $\Theta^*$. Kernel-graph alignment is introduced in this work.

each gradient descent step for GNNs resembles classic *Label Propagation (LP)* algorithms (Szummer & Jaakkola, 2001; Zhu & Ghahramani, 2002; Zhu et al., 2003; Zhou et al., 2003; Chapelle et al., 2009) that propagate true labels from the training set to unseen samples along the graph, as illustrated in Fig. 1(a). Such a characterization of the evolution of learned GNN functions provides answers to question ① and reveals the role of graphs in biasing this procedure.

∘ ***Rethinking Generalization and Heterophily:*** The alignment between NTK $\Theta_t$ and graph adjacency $\mathbf{A}$ (dubbed as kernel-graph alignment) explains success and pathology of learned GNN functions. Let $\Theta^*$ be the optimal kernel denoting if samples share the same (true) label, and the alignment between $\Theta^*$ and $\mathbf{A}$ indicates the homophily level of a graph (i.e. whether connected nodes indeed have the same label). As illustrated by the triangle relation in Fig. 1(b), a larger homophily level adding inherently good kernel-graph alignment translates to better kernel-target alignment (Cristianini et al., 2001), which is desired for favorable generalization. This answers question ② and sheds additional insights on heterophily problems. As support, we establish a strong correlation between generalization and homophily level in overparameterized regimes, and characterize the Bayesian optimality of GNNs that minimizes the population risk. Our empirical results on the time evolution of real-world GNN NTKs on various datasets serves to corroborate our analysis.

**Practical Contributions.** Drawing upon the novel message passing perspective on the learning dynamics of neural networks, we propose to directly replace the NTK matrix with the adjacency matrix $\mathbf{A}$, which leads to a parameter-free algorithm which we refer to as *Residual Propagation (RP)*. Theoretically, it serves as an extreme-case characterization of the learning dynamics of GNNs where graph structure dominates the optimization process. Empirically, we found this embarrassingly simple algorithm can outperform GNNs by $3.0\%$ on OGB benchmarks even without using node features (improvement also shown on 12 additional datasets in Appendix). And notably RP takes orders of magnitude ($< 0.01$) less time to excel the performance of GNNs.

## 2   Preliminaries

**Notations and Setup.** Given a set of training inputs $\mathcal{X} = \{\boldsymbol{x}_i\}_{i=1}^{n_l} \in \mathbb{R}^{n_l \times d}$ and labels $\mathcal{Y} = \{y_i\}_{i=1}^{n_l} \in \mathbb{R}^{n_l}$, where $n_l$ is the size of the labeled instances, we aim to learn a predictive function $f(\boldsymbol{x})$ (such as a neural network) parameterized by weights $\mathbf{W}$. For (semi-)supervised learning, we minimize the squared loss $\mathcal{L}$ using *Gradient Descent (GD)*,

$$\mathcal{L} = \|\mathcal{F}_t - \mathcal{Y}\|^2/2, \quad \partial \mathbf{W}_t/\partial t = -\eta \nabla_{\mathbf{W}} \mathcal{L}, \tag{1}$$

where $\mathbf{W}_t$ and $\mathcal{F}_t = \{f_t(\boldsymbol{x})\}_{\boldsymbol{x} \in \mathcal{X}} \in \mathbb{R}^{n_l}$ are weights and predictions for the training set at optimization time index $t$, and $\eta$ is the learning rate. Temporal discretization of this gradient flow system with time-step $\Delta t = 1$ yields the fixed step-size GD algorithm commonly used in practice, i.e. $\mathbf{W}_{t+1} = \mathbf{W}_t - \eta \nabla_{\mathbf{W}} \mathcal{L}$. Additionally, $\mathcal{X}'$ and $\mathcal{Y}'$ denote testing instances, such that $\bar{\mathcal{X}} = [\mathcal{X}, \mathcal{X}'] \in \mathbb{R}^{n \times d}$ (resp. $\bar{\mathcal{Y}}$) represents the concatenation of training and testing inputs (resp. labels), where $n$ is the full dataset size. For convenience, we generally refer to $f_t(\boldsymbol{x})$ as prediction for a single data point, which is allowed to use additional information such as input features of other nodes by slightly abusing notation, $\mathcal{F}_t$ and $\mathcal{F}'_t$ as predictions for the training and testing sets, which are agnostic to model parameterization in order to apply to parameter-free models. Our analysis generalizes to other loss functions and multi-dimensional output (see Appendix. C.2 and C.3).

Following (Xu et al., 2021a), we focus on node classification or regression tasks, where instances (i.e. nodes) and their relations (i.e. edges) are described by an undirected graph $\mathcal{G} = (\mathcal{V}, \mathcal{E})$, $|\mathcal{V}| = n$. The graph defines a symmetric adjacency matrix $\mathbf{A} \in \mathbb{R}^{n \times n}$ where $\mathbf{A}_{ij} = \mathbf{A}_{\boldsymbol{x}_i \boldsymbol{x}_j} = 1$ for a pair of connected nodes $(\boldsymbol{x}_i, \boldsymbol{x}_j)$ otherwise 0. Based on the data split, we denote submatrices of $\mathbf{A}$ using $\mathbf{A}_{\mathcal{X}\mathcal{X}}$ and $\mathbf{A}_{\mathcal{X}'\mathcal{X}}$. As a standard technique in graph theory (Chung, 1997), we let $\mathbf{A}$ be normalized, i.e. $\mathbf{A} \leftarrow \mathbf{D}^{-\frac{1}{2}}(\mathbf{A} + \mathbf{I})\mathbf{D}^{-\frac{1}{2}}$ where $\mathbf{D}$ is the node degree matrix. Our insights apply to both transductive and inductive settings (see Appendix. C.1), but will focus on the former case unless stated otherwise.

**Graph Neural Networks (GNNs)** are a class of network architectures for learning representations on graphs, commonly trained based on the aforementioned optimization procedure. For GCN (Kipf & Welling, 2017) (and other GNNs that differ by the definition of $\mathbf{A}$), each layer can be written as

$$\mathbf{Z}^{(\ell)} = \sigma(\mathbf{A}\mathbf{Z}^{(\ell-1)}\mathbf{W}^{(\ell)}) \in \mathbb{R}^{n \times m}, \tag{2}$$

where $\mathbf{Z}^{(\ell)}$ are node representations at the $\ell$-th layer with $\mathbf{Z}^{(0)} = \bar{\mathcal{X}}$, $\sigma$ is ReLU activation, and $m$ is the model width. We denote the GNN prediction for a single data point as $f(\boldsymbol{x}; \mathbf{A})$, conditioned on $\mathbf{A}$ to distinguish it from other models. We adopt this architecture as the default GNN model.

**Label Propagation (LP)** represents a class of algorithms for semi-supervised learning, ground-truth labels $\mathcal{Y}$ are propagated along edges to predict $\mathcal{F}'$ for the testing (unlabeled) set. From (Zhu & Ghahramani, 2002), the LP update equation can be written as

$$\text{LP}(\mathcal{Y}; k, \alpha) = [\mathcal{F}_k, \mathcal{F}'_k] = \alpha \mathbf{A}[\mathcal{F}_{k-1}, \mathcal{F}'_{k-1}] + (1 - \alpha)[\mathcal{Y}, \mathbf{0}], \tag{3}$$

where $[\cdot, \cdot]$ is concatenation, $k$ is the iteration number, and $\alpha$ is a hyperparameter. As initialization $[\mathcal{F}_0, \mathcal{F}'_0] = [\mathcal{Y}, \mathbf{0}]$, and after convergence $[\mathcal{F}_\infty, \mathcal{F}'_\infty] \propto (\mathbf{I}_n - \alpha\mathbf{A})^{-1}[\mathcal{Y}, \mathbf{0}]$. LP algorithms have found wide applicability due to their superior efficiency, scalability, and simplicity for deployment.

## 3 A LABEL PROPAGATION VIEW OF GRADIENT DESCENT DYNAMICS

Before delving into GNNs specifically, we detour by providing a LP perspective on the evolution of general parameterized models during GD-based optimization. From this vantage point we propose a simple and interpretable propagation algorithm for semi-supervised learning, with an aim towards illustrating core ideas that will later contribute to our understanding of GNN behavior during optimization (Sec. 3.1). This new algorithm has interesting connections with classic LP algorithms, encompassing them as special cases with additional global optimization guarantees. We evaluate on `OGB` benchmarks (and additional 12 datasets in Appendix E) to verify effectiveness, efficiency, and scalability (Sec. 3.2). We will circle back to connections with GNN learning dynamics in Section 4.

### 3.1 LEARNING DYNAMICS IN FUNCTION SPACE AND RESIDUAL PROPAGATION

We start by characterizing the evolution of a general model (with no restriction on model architecture) in its function space $\mathcal{F}_t$, induced by GD-based optimization that continuously updates the weights $\mathbf{W}_t$. On the training set, this process can be described by (Jacot et al., 2018)

$$\partial \mathcal{F}_t / \partial t = \eta \, \boldsymbol{\Theta}_t(\mathcal{X}, \mathcal{X})\mathcal{R}_t, \quad \text{(NTK Matrix): } \boldsymbol{\Theta}_t(\mathcal{X}, \mathcal{X}) \triangleq \nabla_{\mathbf{W}}\mathcal{F}_t^\top \nabla_{\mathbf{W}}\mathcal{F}_t \in \mathbb{R}^{n_l \times n_l}, \tag{4}$$

where $\mathcal{R}_t = \mathcal{Y} - \mathcal{F}_t \in \mathbb{R}^{n_l}$ denotes *residuals* (Hastie et al., 2009) (a.k.a. errors), the difference between ground-truth labels and model predictions. The so-called *Neural Tangent Kernel (NTK)* (Jacot et al., 2018) $\boldsymbol{\Theta}_t(\mathcal{X}, \mathcal{X})$ is produced by the product between Jacobian matrices, which is dependent on the network architecture, and evolves over time due to its association with time-varying weights $\mathbf{W}_t$. Specially, if the kernel is constant (such as for linear models and infinitely-wide neural networks), (4) reduces to the learning dynamics of kernel regression (Shawe-Taylor & Cristianini, 2004).

**Information Propagation Interpretation.** While (4) has found usage for analyzing the convergence of empirical risk (Du et al., 2019a; Arora et al., 2019) and the spectral bias of deep learning (Mei et al., 2019; Cao et al., 2019), it is restricted to a limited set of samples (i.e. training set). To see how the model evolves on arbitrary inputs for fully characterizing the learned function, we extend (4) to accommodate unseen samples (which could be chosen arbitrarily). Specifically, let $\mathcal{R}'_t = \mathcal{Y}' - \mathcal{F}'_t$ denote residuals for the testing set, and after temporal discretization, (4) can be extended and rewritten neatly using a single variable residual $\mathcal{R}$ (see derivation in Appendix B.1)

$$[\mathcal{R}_{t+1}, \mathcal{R}'_{t+1}] = -\eta \begin{bmatrix} \boldsymbol{\Theta}_t(\mathcal{X}, \mathcal{X}) & \boldsymbol{\Theta}_t(\mathcal{X}, \mathcal{X}') \\ \boldsymbol{\Theta}_t(\mathcal{X}', \mathcal{X}) & \boldsymbol{\Theta}_t(\mathcal{X}', \mathcal{X}') \end{bmatrix} [\mathcal{R}_t, \mathbf{0}] + [\mathcal{R}_t, \mathcal{R}'_t], \tag{5}$$

where $\boldsymbol{\Theta}_t(\mathcal{X}', \mathcal{X}) \triangleq \nabla_{\mathbf{W}} \mathcal{F}_t'^\top \nabla_{\mathbf{W}} \mathcal{F}_t \in \mathbb{R}^{(n-n_l) \times n_l}$ is the NTK matrix between training and testing sets, which quantifies similarity between seen and unseen instances based on how differently their outputs change by an infinitesimal perturbation of weights. The whole $n \times n$ NTK matrix will be henceforth abbreviated as $\boldsymbol{\Theta}_t$. This equation implies that the GD-based optimization of arbitrary parameterized models can be viewed as an information propagation process where residuals propagate unidirectionally from training to arbitrary unseen samples based on a kernel similarity measure.

**Residual Propagation.** Drawing upon an analogy between the information propagation process in (5) induced by optimization, and message passing schemes between instances commonly seen in graph learning, we propose to replace the dense NTK matrix $\boldsymbol{\Theta}_t$ with a sparse matrix (e.g. high-order graph adjacency matrix $\mathbf{A}^K$) as a similarity measure, which allows us to implement (5) as a new semi-supervised algorithm called *Residual Propagation (RP)* with update equation:

$$\left[ \mathcal{R}_{t+1}, \mathcal{R}_{t+1}' \right] = -\eta \mathbf{A}^K [\mathcal{R}_t, \mathbf{0}] + [\mathcal{R}_t, \mathcal{R}_t'], \quad \text{where } \mathcal{R}_0 = \mathcal{Y}, \mathcal{R}_0' = \mathbf{0}. \tag{6}$$

As initialization, the model predictions $\mathcal{F}_0$ and $\mathcal{F}_0'$ are defined as $\mathbf{0}$, and unknown testing labels are defined as $\mathcal{Y}' = \mathbf{0}$ such that negative $\mathcal{R}_t'$ equals model predictions. Compared with (5) merely as an expression, RP is a practical algorithm without trainable weights or relying on input features, and is efficient with linear time complexity owing to sparse matrix multiplication. For further intuition, consider an arbitrary unseen data point $\boldsymbol{x}'$, whose update equation in RP is

**Direction of (ground-truth) label propagation**

$$\boxed{f_{t+1}(\boldsymbol{x}')} = f_t(\boldsymbol{x}') + \eta \sum_{\boldsymbol{x} \in \mathcal{X}} \mathbf{A}_{\boldsymbol{x}'\boldsymbol{x}}^K \left( \boxed{y(\boldsymbol{x})} - f_t(\boldsymbol{x}) \right), \tag{7}$$

where $y(\boldsymbol{x})$ is the ground-truth label for $\boldsymbol{x}$, and $\mathbf{A}_{\boldsymbol{x}'\boldsymbol{x}}^K$ is an element of $\mathbf{A}^K$ corresponding to the sample pair $(\boldsymbol{x}, \boldsymbol{x}')$. For an unseen instance $\boldsymbol{x}'$ that is more 'similar' or 'close' to $\boldsymbol{x}$, more ground-truth label information $y(\boldsymbol{x})$ will then propagate to the testing instance, and vice versa, to shape the prediction for the unseen one (illustrated in Fig. 1(a)). For $f_t(\boldsymbol{x}) \neq 0$, the ground-truth label is adjusted by subtracting current model prediction, i.e. $y(\boldsymbol{x}) - f_t(\boldsymbol{x})$, enabling the propagation process to diminish progressively as the predictions become more accurate.

## 3.2 THEORETICAL ANALYSIS AND EMPIRICAL EVALUATION

Intriguingly, we found the RP algorithm is connected with various classic methods including LP and kernel regression, though they emerge from very different contexts. (Analysis here for RP is also useful for understanding GNNs as will be discussed in the next section.)

**Proposition 1** (Connection with Label Propagation). *The first step of RP in (6) yields identical classification results as LP in (3) (with $\alpha = 1$ and $k = K$):*

$$\textit{(First Step of RP): } [\mathcal{F}_1, \mathcal{F}_1'] = \eta \mathbf{A}^K[\mathcal{Y}, \mathbf{0}], \quad \textit{(Label Propagation): } \mathrm{LP}(\mathcal{Y}; K, 1) = \mathbf{A}^K[\mathcal{Y}, \mathbf{0}]. \tag{8}$$

*Remark.* Besides the first step, each of subsequent step of RP can also be viewed as LP on adjusted ground-truth labels, i.e. $\mathcal{Y} - \mathcal{F}_t = \mathcal{R}_t$. This connection offers the potential for improvement as we iterate beyond the initial LP solution, and also demonstrates the flexibility of RP by replacing $\mathbf{A}^K$ with a more general function $\mathrm{LP}^*(\cdot) : \mathbb{R}^{n_l} \to \mathbb{R}^n$ that takes as input ground-truth labels and outputs predictions, leading to generalized RP with the following update equation:

$$\left[ \mathcal{R}_{t+1}, \mathcal{R}_{t+1}' \right] = -\eta \, \mathrm{LP}^*(\mathcal{R}_t) + [\mathcal{R}_t, \mathcal{R}_t']. \tag{9}$$

**Theorem 2** (Convergence and Optimization Guarantee). *For RP in (6) and sufficiently small step size $\eta < 2\sigma_{max}^{-1}[\mathbf{A}_{\mathcal{X}\mathcal{X}}^K]$, where $\mathbf{A}_{\mathcal{X}\mathcal{X}}^K$ is a submatrix of $\mathbf{A}^K$, and $\sigma_{max}[\mathbf{A}_{\mathcal{X}\mathcal{X}}^K]$ is its largest eigenvalue, $\mathcal{R}_t$ and $\mathcal{R}_t'$ converge as $t \to \infty$ for positive definite $\mathbf{A}_{\mathcal{X}\mathcal{X}}^K$ or positive semi-definite $\mathbf{A}^K$.*

**Corollary 3** (Connection with Kernel Regression). *Upon convergence for positive definite $\mathbf{A}_{\mathcal{X}\mathcal{X}}^K$, the model predictions are equivalent to the kernel regression solution w.r.t. kernel $\kappa(\boldsymbol{x}, \boldsymbol{x}') \triangleq \mathbf{A}_{\boldsymbol{x}\boldsymbol{x}'}^K$*

$$\mathcal{F}_\infty = \mathcal{Y} \quad \textit{(Perfect fit of training set)}, \quad \mathcal{F}_\infty' = \mathbf{A}_{\mathcal{X}'\mathcal{X}}^K (\mathbf{A}_{\mathcal{X}\mathcal{X}}^K)^{-1} \mathcal{Y}. \tag{10}$$

*Remark.* There exists one exception though where the convergence is not guaranteed, namely when $\mathbf{A}$ has negative eigenvalues and $K$ is odd. While this can be easily avoided by adding diagonal components (e.g. PageRank matrix $\alpha \mathbf{A} + (1 - \alpha)\mathbf{I}_n$) for enforcing positive semi-definiteness, empirically we do not need the algorithm to converge by stopping at the step with peak validation performance (just like standard training procedure for neural networks), and the algorithm can still achieve satisfactory generalization performance (often better than the version of RP that can converge).

Table 1: Empirical evaluation of RP on `OGB` datasets. Accuracy is reported for `Arxiv` and `Products`, and ROC-AUC for `Proteins`. Last three rows compare RP against full-batch GNN.

| Model | Feat. | Arxiv | | Proteins | | Products | | Avg. | # Param. |
| | | Validation | Test | Validation | Test | Validation | Test | | |
|---|---|---|---|---|---|---|---|---|---|
| MLP | $\mathcal{X}$ | $57.65 \pm 0.12$ | $55.50 \pm 0.23$ | $77.06 \pm 0.14$ | $72.04 \pm 0.48$ | $75.54 \pm 0.14$ | $61.06 \pm 0.08$ | 66.48 | $O(\ell m^2)$ |
| LinearGNN | $\mathcal{X}, \mathbf{A}$ | $70.67 \pm 0.02$ | $69.39 \pm 0.11$ | $66.11 \pm 0.87$ | $62.89 \pm 0.11$ | $88.97 \pm 0.01$ | $74.21 \pm 0.04$ | 72.04 | $O(dc)$ |
| GNN | $\mathcal{X}, \mathbf{A}$ | $73.00 \pm 0.17$ | $71.74 \pm 0.29$ | $79.21 \pm 0.18$ | $72.51 \pm 0.35$ | $92.00 \pm 0.03$ | $75.64 \pm 0.21$ | 77.35 | $O(\ell m^2)$ |
| LP | $\mathbf{A}$ | $70.14 \pm 0.00$ | $68.32 \pm 0.00$ | $83.02 \pm 0.00$ | $74.73 \pm 0.00$ | $90.91 \pm 0.00$ | $74.34 \pm 0.00$ | 76.91 | 0 |
| RP (ours) | $\mathbf{A}$ | $71.37 \pm 0.00$ | $70.06 \pm 0.00$ | $85.19 \pm 0.00$ | $78.17 \pm 0.00$ | $91.31 \pm 0.00$ | $78.25 \pm 0.00$ | 79.06 | 0 |
| Speedup / step | | $\times$ 14.48 | | $\times$ 14.00 | | $\times$ 12.46 | | | |
| Time to Acc. | | $\times$ 0.01461 | | $\times$ 0.00008 | | $\times$ 0.00427 | | | |
| Memory | | $\times$ 0.094 | | $\times$ 0.363 | | $\times$ 0.151 | | | |

Figure 2: Learning curves of RP and comparison with the performance of LP ($\alpha = 1$), linear GNN and deep GNN. Transition from yellow to purple denotes RP with decreasing step size $\eta$.

**Empirical Evaluation.** We compare the basic version of RP from (6) with some representative GNN architectures (LinearGNN Wu et al. (2019) and GCN Kipf & Welling (2017)) on challenging `OGB` (Hu et al., 2020) node classification datasets `Arxiv`, `Proteins`, `Products` with up to millions of nodes and edges. Details of experimental setting are deferred to Appendix D.1.

○ *Effectiveness:* In Table 1, the proposed RP demonstrates decent performance, often surpassing GNNs even when no node or edge feature information is utilized. Notably, this performance even approaches some other more advanced models such as node-level graph Transformers (Wu et al., 2022b). As depicted in Fig. 2, RP achieves the same performance as LP using one step, and quickly increases until reaching its peak performance, which surpasses the GNN. Specifically, on `Proteins` where the graph contains relatively richer structural information, a single step of RP exceeds a well-trained deep GNN, while in `Products`, 4 steps of RP exceeds the GNN.

○ *Efficiency and Scalability:* In addition to its effectiveness, RP offers several advantages. It does not require learnable parameters and boosts speed, with each step being more than 10 times faster than the GNN, in part because it does not need a backward pass typically required in conventional learning frameworks. Furthermore, RP achieves its peak performance within a mere 20 steps, and thus overall takes orders of magnitudes less time ($< 0.01$) to attain GNN-level performance. In terms of memory cost, RP inherits the scalability of LP and only requires storage space for model predictions; in practice, we often found the memory bottleneck lies with data preprocessing $O(|\mathcal{E}|)$ rather than the algorithm itself $O(n)$.

In Appendices E.1 and E.2 respectively, we discuss extensions of RP that combine with kernels (e.g. Gaussian kernel) to accommodate node features, and test additional 7 datasets (`Cora`, `Citeseer`, `Pubmed`, `Computer`, `Photo`, `CS`, `Physics`) where RP can still outperform popular GNNs.

# 4 LEARNING DYNAMICS OF GRAPH NEURAL NETWORKS

We have introduced a simple propagation method that competes with deep GNN models, which raises the question whether or not GNNs are also (secretly) taking advantage of similar residual propagating effects with a graph inductive bias in their optimization process. To examine this possibility, we formally study the learning dynamics of GNNs in function space during GD-based training.

## 4.1 GENERAL FORM BY NODE-LEVEL GNTK

Similar to the learning dynamics of general parameterized models in (5), the learning dynamics of GNNs in node-level tasks is characterized by their NTK defined as follows:

**Definition 4** (Node-Level Graph Neural Tangent Kernel). *For a $\ell$-layer GNN in node-level tasks defined in Sec. 2, the NTK is defined as*

$$\Theta_t^{(\ell)}(\boldsymbol{x}, \boldsymbol{x}'; \mathbf{A}) = \nabla_{\mathbf{W}} f(\boldsymbol{x}; \mathbf{A})^\top \nabla_{\mathbf{W}} f(\boldsymbol{x}'; \mathbf{A}), \tag{11}$$

*which we refer to as Node-Level Graph Neural Tangent Kernel (GNTK) to differentiate it with the graph-level GNTK for graph-level tasks proposed in Du et al. (2019b), or simply NTK of GNNs.*

How GNNs evolve in function space during training also follows the residual propagation process:

$$\left[\mathcal{R}_{t+1}, \mathcal{R}'_{t+1}\right] = -\eta \boldsymbol{\Theta}_t^{(\ell)}[\mathcal{R}_t, \mathbf{0}] + [\mathcal{R}_t, \mathcal{R}'_t], \quad \boldsymbol{\Theta}_t^{(\ell)} \triangleq [\boldsymbol{\Theta}_t^{(\ell)}(\boldsymbol{x}_i, \boldsymbol{x}_j; \mathbf{A})]_{i,j \in [n] \times [n]}, \quad (12)$$

where the initial values of residuals depend on initial weights. A precise characterization of (12) requires mathematically computing the node-level GNTK, which is prohibitively challenging for finitely-wide GNNs. Fortunately, we can derive its explicit formula in overparameterized regimes where the model width $m$ tends to infinity, following that of fully-connected neural networks (Jacot et al., 2018; Lee et al., 2019). Given that the limiting NTK in this regime behaves as a constant kernel function (Yang & Littwin, 2021), we change the notation by omitting the subscript $t$.

Concretely, we present the computation of the node-level GNTK in a recurrent layer-wise manner where each layer is decomposed into two steps: *Feature Transformation* (corresp. $\mathbf{Z} \leftarrow \sigma(\mathbf{ZW})$) and *Feature Propagation* (corresp. $\mathbf{Z} \leftarrow \mathbf{AZ}$). As elements in the computation, we denote a node-level GNTK for GNN without feature propagation at $\ell$-th layer as $\bar{\boldsymbol{\Theta}}^{(\ell)}$, the covariance matrix of the $\ell$-th layer's outputs with (without) feature propagation as $\boldsymbol{\Sigma}^{(\ell)}$ ($\bar{\boldsymbol{\Sigma}}^{(\ell)}$), and the covariance matrix of the derivative to the $\ell$-th layer as $\dot{\boldsymbol{\Sigma}}^{(\ell)}$. Then the feature propagation and transformation steps in each layer respectively correspond to (we concisely show the key steps here and defer the complete formula to Appendix B.3):

$$\begin{array}{cc} (\textit{Feature Transformation}) & (\textit{Feature Propagation}) \\[4pt] \bar{\boldsymbol{\Theta}}^{(\ell)} = \boldsymbol{\Theta}^{(\ell-1)} \odot \dot{\boldsymbol{\Sigma}}^{(\ell)} + \bar{\boldsymbol{\Sigma}}^{(\ell)}, & \left\{ \begin{array}{l} \boldsymbol{\Sigma}^{(\ell)} = \mathbf{A}\, \bar{\boldsymbol{\Sigma}}^{(\ell)} \mathbf{A} \\ \boldsymbol{\Theta}^{(\ell)} = \mathbf{A}\, \bar{\boldsymbol{\Theta}}^{(\ell)} \mathbf{A}. \end{array} \right. \end{array} \quad (13)$$

**Implications.** Compared with the NTK computation for a fully-connected neural network (Jacot et al., 2018), the node-level GNTK has an equivalent feature transformation step, while its uniqueness stems from the feature propagation step, whereby the adjacency matrix $\mathbf{A}$ integrates into the kernel similarity measure. Consequently, this kernel function naturally accommodates a graph inductive bias, and thus the residual propagation process of GNNs also tends to follow the trajectory regulated by the graph, similar to the behavior of the RP algorithm. While such an analysis offers valuable mathematical intuitions, we next present how the learning dynamics of certain GNNs in function space can be exactly formulated under the framework of generalized RP.

### 4.2 CASE STUDIES: FROM TWO LAYERS TO ARBITRARY DEPTH

Given that our primary focus centers on the role of graphs in GNNs (as without them, GNNs are largely equivalent to MLPs), we exclude external node features and instead define inputs as either: 1) an identity matrix $\bar{\mathcal{X}} \triangleq \mathbf{I}_n$ that assigns each node a one-hot vector as indication of its unique identity (as sometimes assumed in practice (Kipf & Welling, 2017; Zhu et al., 2021)), which can be viewed as learning a unique embedding for each node by treating the first-layer weights as an embedding table; 2) fixed node embeddings from graph spectral decomposition $\bar{\mathcal{X}} \triangleq \arg\min_{\mathbf{B}} \|\mathbf{A} - \mathbf{BB}^\top\|_F^2$, which aligns with various network embedding approaches based on definitions of $\mathbf{A}$ (Qiu et al., 2018).

**Two-Layer GNN.** Building on the setup from prior work analyzing infinitely-wide two-layer neural networks (Arora et al., 2019), we consider two-layer GNNs:

**Theorem 5** (Two-Layer GNN Dynamics). *For an infinitely-wide two-layer GNN defined as $[\mathcal{F}, \mathcal{F}'] = \mathbf{A}\sigma(\mathbf{A}\bar{\mathcal{X}}\mathbf{W}^{(1)})\mathbf{W}^{(2)}/\sqrt{m}$ with ReLU activation, $\bar{\mathcal{X}} = \mathbf{I}_n$ and standard NTK parameterization, its learning dynamics by optimizing $\mathbf{W}^{(1)}$ can be written as a generalized RP process*

$$\left[\mathcal{R}_{t+1}, \mathcal{R}'_{t+1}\right] = -\eta \mathbf{A}(\mathbf{A}^2 \odot \mathbf{S})\mathbf{A}[\mathcal{R}_t, \mathbf{0}] + [\mathcal{R}_t, \mathcal{R}'_t], \quad \mathbf{S}_{ij} = \left(\pi - \arccos(\frac{\mathbf{A}_i^\top \mathbf{A}_j}{\|\mathbf{A}_i\|\|\mathbf{A}_j\|})\right)/2\pi. \quad (14)$$

*The matrix $\mathbf{S}$ reweights each element in $\mathbf{A}^2$ by the similarity of neighborhood distributions of two nodes. For $\bar{\mathcal{X}} = \arg\min_{\mathbf{B}} \|\mathbf{A} - \mathbf{BB}^\top\|_F^2$, the propagation matrix is replaced by $\mathbf{A}(\mathbf{A}^3 \odot \tilde{\mathbf{S}})\mathbf{A}$ where $\tilde{\mathbf{S}}$ is another reweighting matrix (details given in Appendix B.4).*

**Arbitrarily Deep GNN.** Pushing further, we can also characterize the evolution of arbitrarily deep GNNs where feature propagation is applied at the last layer (e.g. Klicpera et al. (2019); Liu et al. (2020); Spinelli et al. (2020); Chien et al. (2021), called decoupled GNNs in some other literature):

**Theorem 6** (Deep and Wide GNN Dynamics). *For arbitrarily deep and infinitely-wide GNNs with feature propagation deferred to the last layer, i.e. $[\mathcal{F}, \mathcal{F}'] = \mathbf{A}^\ell \text{MLP}(\bar{\mathcal{X}})$ with $\bar{\mathcal{X}} = \mathbf{I}_n$, the learning dynamics that result from optimizing MLP weights can be written as the generalized RP process*

$$\left[\mathcal{R}_{t+1}, \mathcal{R}'_{t+1}\right] = -\eta \mathbf{A}^\ell(\mathbf{I}_n + c\mathbf{1}\mathbf{1}^\top)\mathbf{A}^\ell[\mathcal{R}_t, \mathbf{0}] + [\mathcal{R}_t, \mathcal{R}'_t], \quad (15)$$

*where $c \geq 0$ is a constant determined by the model depth, and $\mathbf{1}$ is an all-1 column vector.*

As a special case of the above, when the backbone MLP only has one layer, the model degrades to a linear GNN (e.g. SGC Wu et al. (2019)), and the constant correspondingly is $c = 0$, giving us the basic version of RP in (6) or its approximation for odd $K$:

**Corollary 7** (Linear GNN Dynamics). *The learning dynamics of the linear GNN $[\mathcal{F}, \mathcal{F}'] = \mathbf{A}^\ell \bar{\mathcal{X}} \mathbf{W}$ is identical to the basic version of RP in (6) with $K = 2\ell$ for input features $\bar{\mathcal{X}} = \mathbf{I}_n$, and $K = 2\ell + 1$ for $\bar{\mathcal{X}} = \arg\min_{\mathbf{B}} \|\mathbf{A} - \mathbf{B}\mathbf{B}^\top\|_F^2$ and positive semi-definite $\mathbf{A}$.*

*Remark.* Despite this equivalence, it is important to note that this specific linear GNN (on our given input features) is not as lightweight as it may appear. First, its parameter number scales with the size of dataset, often reaching orders of magnitude larger than deep GNN models (e.g., $115, 104, 363$ for the linear model v.s. $103, 727$ for GCN on `Products`). Additionally, the full-rank spectral decomposition of $\mathbf{A}$ is computationally very expensive ($O(n^3)$) for large graphs. In contrast, the basic version RP in (6) efficiently yields identical results to this heavily parameterized GNN without actually training parameters or computing expensive matrix decompositions. Furthermore, the generalized RP from (9) can implement algorithms that are infeasible within a conventional deep learning framework, e.g. training infinitely-wide GNNs.

## 5   RETHINKING THE SUCCESS AND PATHOLOGY OF GNNS

In the last section, we have given theoretical analysis with concrete examples to illustrate that NTKs of GNNs naturally tend to align with the graph. This finding, coupled with the consistently strong performance of the RP algorithm — an example of perfect graph-kernel alignment — prompts the question of whether this alignment between NTK and the graph plays a crucial role in understanding the success of GNNs. Next, we offer interpretable explanations of "when and why GNNs successfully generalize" and their pathological training behavior on heterophilic graphs (Sec. 5.1). We also study the time evolution of real-world GNN NTKs to empirically verify our theoretical results (Sec. 5.2).

### 5.1   WHEN AND WHY GNNS SUCCESSFULLY GENERALIZE?

Our previous discussions have revolved around two matrices, namely the graph adjacency matrix $\mathbf{A}$ and the NTK matrix $\mathbf{\Theta}_t$.[1] To complete the theoretical picture, we introduce another matrix called the ideal *or optimal kernel matrix* (Cristianini et al., 2001), defined as $\mathbf{\Theta}^* \triangleq \bar{\mathcal{Y}}\bar{\mathcal{Y}}^\top \in \mathbb{R}^{n \times n}$ to indicate whether two instances have the same label, and a metric to quantify alignment of (kernel) matrices:

**Definition 8** (Alignment, Cristianini et al. (2001)). *Given two (kernel) matrices $\mathbf{K}_1$ and $\mathbf{K}_2$, their alignment is defined as $A(\mathbf{K}_1, \mathbf{K}_2) \triangleq \langle \mathbf{K}_1, \mathbf{K}_2 \rangle_F / (\|\mathbf{K}_1\|_F \|\mathbf{K}_2\|_F) \in [0, 1]$. This is a generalization of cosine similarity from vectors to matrices,* $\arccos$ *of which satisfies the triangle inequality.*

○ *Homophily Level*: $A(\mathbf{A}, \mathbf{\Theta}^*)$. The alignment between $\mathbf{A}$ and $\mathbf{\Theta}^*$ quantifies the *homophily level* of graph structure, i.e. whether two connected nodes indeed have the same label, and is determined the dataset. While many empirical results (e.g. Zhu et al. (2020)) suggest high homophily level is important for the performance of GNNs, deeper theoretical understandings are mostly lacking.

○ *Kernel-Target Alignment*: $A(\mathbf{\Theta}_t, \mathbf{\Theta}^*)$. The alignment between kernel matrix and optimal $\mathbf{\Theta}^*$ has been widely studied in the research on kernel learning (Cristianini et al., 2001; Kwok & Tsang, 2003; Lanckriet et al., 2004; Gönen & Alpaydın, 2011). Better kernel-target alignment has been recognized as a critical factor that leads to favorable generalization. For intuition, one can quickly verify that substituting $\mathbf{\Theta}^*$ to the residual dynamics in (5) leads to perfect generalization performance (since ground-truth labels only propagate to unseen instances with the same label).

○ *Kernel-Graph Alignment*: $A(\mathbf{\Theta}_t, \mathbf{A})$. The alignment between NTK and graph is a novel notion in our work, as prior sections have shown that GNN NTK matrices naturally tend to align with $\mathbf{A}$. E.g., the RP algorithm (and variants thereof) serve as an extreme case with two identical matrices.

**Implications.** We consider two cases. For *homophilic* graphs where $A(\mathbf{A}, \mathbf{\Theta}^*) \uparrow$ is naturally large, better kernel-graph alignment $A(\mathbf{\Theta}_t, \mathbf{A}) \uparrow$ consequently leads to better kernel-target alignment $A(\mathbf{\Theta}_t, \mathbf{\Theta}^*) \uparrow$. In other words, the NTK of GNNs naturally approaches the optimum as the graph structure possesses homophily property, and leveraging it in the optimization process (12) encourages training residuals to flow to unseen samples with the same label and thus better generalization; In contrast, for *heterophilic* graphs where $A(\mathbf{A}, \mathbf{\Theta}^*)$ is small, better kernel-graph alignment will hinder kernel-target alignment, explaining the pathological learning behavior of GNNs when dealing with heterophilic graphs in an interpretable manner.

---

[1]We here refer to $\mathbf{A}$ as a class of similarity matrices based on original $\mathbf{A}$ in a general sense, such as $\mathbf{A}^K$ etc.

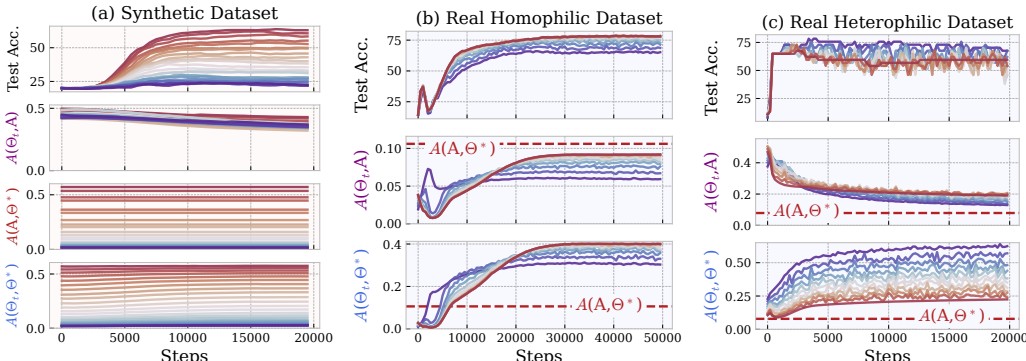

Figure 3: Evolution of NTK matrix $\mathbf{\Theta}_t$ of finitely-wide GCN during training, reflected by matrix alignment. **(a)** Synthetic dataset generated by a stochastic block model, where the homophily level gradually decreases by altering edge probabilities, i.e. homophilic → heterophilic; **(b & c)** Real-world homophilic (Cora) and heterophilic (Texas) datasets, where the graph is gradually coarsened until there is no edge left when evaluating $\mathbf{\Theta}_t$, i.e. more graph → less graph. (Details in Appendix D.2)

**Theoretical Support 1.** To support this interpretation, we examine the generalization behavior of infinitely-wide neural networks in the extreme case where its NTK matrix is assumed to be governed by the graph, say, $\lim_{k\to\infty}\sum_{i=0}^{k}(\alpha\mathbf{A})^i$ with $\alpha\in(0,1)$ as adopted by the converged LP algorithm in (3). With a common assumption that training instances are drawn i.i.d. from a distribution $\mathcal{P}$, and based on the Rademacher complexity generalization bound for kernel regression (Bartlett & Mendelson, 2002; Arora et al., 2019; Du et al., 2019b), we have a label-dependent high-probability (at least $1-\delta$) upper bound on population risk (derivation in Appendix B.7):

$$\mathbb{E}_{(\boldsymbol{x},y)\sim\mathcal{P}}\left[l\left(f(\boldsymbol{x}),y\right)\right] = O\left(\sqrt{\frac{n_l - cA(\mathbf{A},\mathbf{\Theta}^*)}{n_l}} + \sqrt{\frac{\log(1/\delta)}{n_l}}\right), \tag{16}$$

where $c = \alpha\|\mathbf{\Theta}^*\|_F\|\mathbf{A}\|_F$ is a constant, $A(\mathbf{A},\mathbf{\Theta}^*)$ is the homophily level for the training set (with slight abuse of notation). This bound can also be viewed as a theoretical guarantee for the converged (generalized) RP algorithm and clearly demonstrates that: *for $A(\mathbf{\Theta}_t,\mathbf{A})$ fixed as 1, higher level of graph homophily $A(\mathbf{A},\mathbf{\Theta}^*)$ plays a dominant role in better generalization*. The above analysis can also be potentially extended to infinitely-wide GNNs discussed in Sec. 4 with further assumptions on the condition number of its NTK matrix, which will lead to similar (but less straightforward) results.

**Theoretical Support 2.** Pushing further, in a complementary setting to the above, where the objective is to find the optimal a priori kernel for directly minimizing the population risk (but without access to any input features or ground-truth labels), we demonstrate that when homophily assumptions are imposed on the graph, infinitely-wide GNNs and the RP algorithm will yield the optimal kernel regression predictor with the provably best generalization performance:

**Theorem 9** (Bayesian Optimality of GNN). *Assume that the underlying data generation distribution $\mathcal{P}$ is such that the probability of a pair of instances having the same label $P(y(\boldsymbol{x}_i) = y(\boldsymbol{x}_j))$ is proportional to $\mathbf{A}_{ij} - 1/2$. Then the optimal kernel regression predictor that minimizes the population risk with squared loss has kernel matrix $\mathbf{A}$.*

*Remark.* The matrix $\mathbf{A}$ from the above result could vary depending on different assumptions on $P(y(\boldsymbol{x}_i) = y(\boldsymbol{x}_j))$, such as $\mathbf{A}^K$ for the basic version of RP, or $\mathbf{A}\left(\mathbf{A}^2 \odot \mathbf{S}\right)\mathbf{A}$ for infinitely-wide two-layer GNNs in Theorem 5. And ideally, if one has privileged access to labels of all instances, this optimal matrix is exactly the optimal (posterior) kernel $\mathbf{\Theta}^*$ discussed above and perfect generalization will be achieved. This result further verifies our interpretation of generalization by showing that *for $A(\mathbf{A},\mathbf{\Theta}^*)$ fixed to be large, better $A(\mathbf{\Theta}_t,\mathbf{A})$ (e.g. GNN and RP) leads to favorable generalization.*

### 5.2 EMPIRICAL VERIFICATION AND FURTHER INSIGHTS ON HETEROPHILY

We next empirically verify results from Sec. 4 and 5.1 by studying the time evolution of real-world GNN NTKs. Figure 3 plots the results reflected by alignment between $\mathbf{\Theta}_t$, $\mathbf{A}$ and $\mathbf{\Theta}^*$.

**Synthetic Dataset.** To demonstrate the effects of different homophily levels, we use a stochastic block model (Holland et al., 1983) to generate a synthetic dataset with informative node features. As shown in Fig. 3(a), the kernel-graph alignment $A(\mathbf{\Theta}_t,\mathbf{A})$ for GNNs stays at a high level regardless of different graph structures, as a natural result of the network architecture (Sec. 4). Consequently, as

we gradually alter edge probabilities to decrease the homophily level $A(\mathbf{A}, \mathbf{\Theta}^*)$, the kernel-target alignment $A(\mathbf{\Theta}_t, \mathbf{\Theta}^*)$ also decreases and the testing accuracy drops dramatically (Sec. 5.1).

**Real-World Datasets.** On real-world homophilic and heterophilic datasets, we progressively coarsen the graph until the GNN degrades to an MLP, allowing us to analyze the effects of feature propagation in the model. For now, let us first focus on comparing red and blue lines in Fig. 3(b,c). We found the kernel-graph alignment $A(\mathbf{\Theta}_t, \mathbf{A})$ overall decreases with less graph structure, again verifying results in Sec. 4. However, the impact of the graph structure depends on different homophily levels $A(\mathbf{A}, \mathbf{\Theta}^*)$: on the homophilic dataset, more graph structure and better $A(\mathbf{\Theta}_t, \mathbf{A})$ optimize the NTK matrix as reflected by better kernel-target alignment $A(\mathbf{\Theta}_t, \mathbf{\Theta}^*)$, but worsens it on the heterophilic one. This results in distinct generalization behavior of GNNs reflected by the testing accuracy.

**Can and How (G)NNs Handle Heterophily?** Recently, there has been a growing discussion on whether standard GNNs are capable of handling heterophily, with empirical results pointing to diverging conclusions (Zhu et al., 2020; Ma et al., 2022; Luan et al., 2022; Platonov et al., 2023). From the learning dynamics perspective, we add new insights to this debate: ∘ **1)** While we have shown that infinitely-wide GNNs are sub-optimal on heterophilic graphs, MLPs in contrast have no guarantee of kernel-target alignment (since both $A(\mathbf{A}, \mathbf{\Theta}^*)$ and $A(\mathbf{\Theta}_t, \mathbf{A})$ are not well-aligned), indicating that *they could either generalize better or worse without clearcut (dis)advantages, as opposed to the case on homophilic graphs where GNN is provably at an advantage*, which explains the diverse empirical results when comparing GNNs with MLPs in prior work. ∘ **2)** As we now turn to the NTK's time evolution in Fig. 3, an additional phenomenon we found across all datasets is the overall increase of kernel-target alignment $A(\mathbf{\Theta}_t, \mathbf{\Theta}^*)$ during the training process. This indicates that the training process enables real-world GNNs to adjust their NTK's feature space such that the kernel matrix leans towards an ultimately homoplilic graph structure (i.e. $\mathbf{\Theta}^*$) to adapt to heterophilic datasets (and consequently different evolutionary trends of $A(\mathbf{\Theta}_t, \mathbf{A})$ for hemophiliac and heterophilic datasets). Such a phenomenon has also been found for other models in vision tasks (Baratin et al., 2021). Additionally, since our analysis also applies to other forms of $\mathbf{A}$ for feature propagation in GNNs, it could also potentially explain how some specialized models with different definitions of $\mathbf{A}$ (e.g. allowing signed propagation) can mitigate the heterophily issue.

## 6 MORE DISCUSSIONS AND CONCLUSION

**Abridged Related Work.** Most existing work on theoretical aspects of GNNs focuses on representation and generalization (see Jegelka (2022) and references therein) while their optimization properties remains under-explored (Zhang et al., 2020; Xu et al., 2021a; Yadati, 2022). Specifically, existing work in representation (or expressiveness) does not provide answers to what exactly GNN functions are found during the optimization process. For generalization, prior work is insufficient to explain the effects of training, which is widely-recognized as a crucial ingredient, and does not connect to heterophily, a relevant aspect in practice. (See unabridged related work in Appendix A.)

**Applicability.** Our insights apply to both transductive and inductive settings, other loss functions, and multi-dimensional outputs (see Appendix C). The analytical framework could also be extended to other tasks (which are left as future work): for graph classification or regression, the definition of GNTK in (12) should be modified to the graph-level one (Du et al., 2019b); for link prediction and self-supervised learning, the loss function in the derivation of (5) should be adjusted accordingly. A similar message passing process in function space would still hold in these settings. Moreover, the proposed RP is compatible with regression tasks due to minimization of the squared loss.

**Broader Impacts.** Theoretically, our framework of interpreting GD dynamics in function space as a label propagation process across seen and unseen samples could be adapted for dissecting other models and tasks, and the analysis in Sec. 5 exemplifies using kernel learning to justify model architectures. Practically, the proposed RP unlocks the possibility of leveraging a sparse structure in complicated training dynamics, acting as efficient forward-only solutions to implement dynamics in function space which are previously impractical or extremely expansive (Arora et al., 2020). More broadly, it also serves as a bridge of classic graph-based algorithms (Chapelle et al., 2009) and modern deep learning, and echoes recent efforts to develop forward-only learning frameworks (Hinton, 2022).

**Conclusion.** This work represents a preliminary endeavor in formally studying the learning dynamics of GNNs in function space. We reveal the alignment between external graph structure and NTK matrix that controls the evolution of models, which enables deeper understanding of the generalization capability and pathological behavior of GNNs by connecting to the study of heterophily, and leads to a minimalist algorithm with superior efficiency, decent performance and theoretical interpretability.

ETHICS STATEMENT

In this research, we investigate the learning dynamics of GNNs in function space and offer insights to their optimization and generalization behaviors. Our insights can deepen understandings for graph-based deep learning, and do not appear to carry immediate harmful consequences to the best of our knowledge.

REPRODUCIBILITY STATEMENT

The complete proofs of our theoretical results and their assumptions are provided in Appendix B. Extensive implementation details of the experiments from Section 3.2 and Section 5.2 regarding baselines, hyperparameters and datasets are provided in Appendix D.1 and D.2. Detailed descriptions for the implementation of different versions of RP algorithms are given in Appendix D.3. Moreover, we plan to release the code in the near future.

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

CONTENTS

# A    UNABRIDGED RELATED WORK

This section discusses more related works that are not covered in the main text and those related works that are already covered but in greater depth.

## A.1    OPTIMIZATION AND LEARNING DYNAMICS

Towards deeper understanding of the success and limitation of GNNs, many works focus on the representation power of GNNs (Maron et al., 2019; Xu et al., 2019; Oono & Suzuki, 2019; Chen et al., 2019; Dehmamy et al., 2019; Sato et al., 2019; Loukas, 2020). While these works formalize what functions a GNN can possibly represent, they do not provide answers to which specific GNN function will be found during optimization process or whether the learned GNN function will successfully generalize. In contrast, theoretical understandings of the optimization properties of GNNs are scarce. Specifically, (Zhang et al., 2020) prove the global convergence of a one-layer GNN with assumptions on the training algorithm built on tensor initialization and accelerated gradient descent; (Xu et al., 2021a) analyze the convergence rate of linearized GNNs with focus on learning dynamics in weight space and empirical risk; (Yadati, 2022) analyze optimization properties of a two-layer GCN by introducing a convex program. However, none of existing studies study the learning dynamics of GNNs in function space, which is our focus and could lead to a wealth of new insights that are theoretically and practically valuable. Recently a surge of works (Yang et al., 2021; Di Giovanni et al., 2022; Wu et al., 2023) connect the gradient dynamics of certain objectives to the architecture of GNNs, which are orthogonal to our contributions.

## A.2    IN- AND OUT-OF-DISTRIBUTION GENERALIZATION

Existing works in generalization of GNNs focus on their in-distribution (Scarselli et al., 2018; Verma & Zhang, 2019; Du et al., 2019b; Liao et al., 2020) and out-of-distribution generalization properties (Yehudai et al., 2021; Xu et al., 2021b; Wu et al., 2022a; Yang et al., 2023). For *node-level tasks* which are challenging due to the dependency between samples, most prior art study the generalization bound of GNNs based on complexity of model class (Scarselli et al., 2018; Baranwal et al., 2021; Garg et al., 2020; Ma et al., 2021) or algorithmic stability (Verma & Zhang, 2019; Zhou & Wang, 2021; Cong et al., 2021). The former line of works do not consider the optimization, which however is a critical ingredient of finding generalizable solutions; the latter line of works consider the training algorithm, but their bounds as the number of epochs becomes large. Moreover, there is no existing work (to the best of our knowledge) formally connecting generalization and heterophily, though their connections are tacitly implied in many empirical results, e.g. (Zhu et al., 2020; 2021; Zheng et al., 2022). The most related work in GNN generalization is (Yang et al., 2023) wherein the authors found vanilla MLPs with test-time message passing operations can be as competitive as different GNN counterparts, and then analyzed generalization of GNNs using node-level GNTK with attention to feature-wise extrapolation. Our results agree with and complement (Yang et al., 2023) by noting that our analysis is applicable for all inductive, transductive and training without graph settings (cf. Appendix C.1).

## A.3    GRAPH-BASED SEMI-SUPERVISED LEARNING (LABEL PROPAGATION)

One of most popular type of semi-supervised learning methods is graph-based methods (Chapelle et al., 2009), where label propagation (Szummer & Jaakkola, 2001; Zhu & Ghahramani, 2002; Zhu et al., 2003; Zhou et al., 2003; Chapelle et al., 2009; Koutra et al., 2011; Gatterbauer et al., 2015; Yamaguchi et al., 2016; Iscen et al., 2019) is one of the most classic and widely-used algorithms. The algorithm relies on external data structure, usually represented by a graph adjacency matrix $\mathbf{A} \in \{0,1\}^{n \times n}$, to propagate ground-truth labels of labeled samples to infer unlabeled ones, and it is still under active research nowadays, e.g. (Pukdee et al., 2023; Lee et al., 2022), mainly due to its efficiency and scalability. The label propagation algorithm can usually be induced from minimization of a quadratic objective (Zhou et al., 2003):

$$\mathcal{E} = \|\mathcal{F} - \mathcal{Y}\|^2 + \lambda \operatorname{Tr} \left[ [\mathcal{F}, \mathcal{F}']^\top (\mathbf{I}_n - \mathbf{A}) [\mathcal{F}, \mathcal{F}'] \right], \tag{17}$$

motivated from the Dirichlet energy to enforce smoothness of predictions according to sample relations. Notice the minima of this objective does not minimize the squared loss part $\|\mathcal{F} - \mathcal{Y}\|^2$

due to the regularization of the second term. In contrast, the proposed RP algorithm has similar smoothness effects but minimizes the squared loss. Moreover, some previous works attempt to explore the interconnections between LP and GNNs from different perspectives such as feature/label influence (Wang & Leskovec, 2021), generative model (Jia & Benson, 2022). Compared with them, we first show the exact equivalence of classification results between LP and the first step training of GNN (on node embeddings and with squared loss).

## B  Theoretical Results

### B.1  Derivation of equation 5: Learning Dynamics of General Model

Recall that for supervised learning, one is interested in minimizing the squared error $\mathcal{L}$ using *gradient descent (GD)*,

$$\mathcal{L} = \frac{1}{2}\|\mathcal{F}_t - \mathcal{Y}\|^2, \quad \frac{\partial \mathbf{W}_t}{\partial t} = -\eta \nabla_{\mathbf{W}} \mathcal{L}. \tag{18}$$

Let $\mathcal{R}_t = \mathcal{Y} - \mathcal{F}_t$ and $\mathcal{R}'_t = \mathcal{Y}' - \mathcal{F}'_t$ denote residuals for the labeled and unlabeled sets. Then, their joint dynamics induced by GD training is

$$\frac{\partial [\mathcal{R}_t, \mathcal{R}'_t]}{\partial t} \tag{19}$$

$$= \frac{\partial [\mathcal{R}_t, \mathcal{R}'_t]}{\partial \mathbf{W}_t} \frac{\partial \mathbf{W}_t}{\partial t} \qquad \text{(Chain rule)} \tag{20}$$

$$= -\eta \frac{\partial [\mathcal{R}_t, \mathcal{R}'_t]}{\partial \mathbf{W}_t} \nabla_{\mathbf{W}} \mathcal{L} \qquad \text{(GD training)} \tag{21}$$

$$= -\eta \frac{\partial [\mathcal{R}_t, \mathcal{R}'_t]}{\partial \mathbf{W}_t} \nabla_{\mathbf{W}} [\mathcal{F}_t, \mathcal{F}'_t] \nabla_{[\mathcal{F}_t, \mathcal{F}'_t]} \mathcal{L} \qquad \text{(Chain rule)} \tag{22}$$

$$= \eta \nabla_{\mathbf{W}} [\mathcal{F}_t, \mathcal{F}'_t]^\top \nabla_{\mathbf{W}} [\mathcal{F}_t, \mathcal{F}'_t] \nabla_{[\mathcal{F}_t, \mathcal{F}'_t]} \mathcal{L} \qquad \text{(Change of notation)} \tag{23}$$

$$= \eta \, \boldsymbol{\Theta}_t([\mathcal{X}, \mathcal{X}'], [\mathcal{X}, \mathcal{X}']) \nabla_{[\mathcal{F}_t, \mathcal{F}'_t]} \mathcal{L} \qquad \text{(Neural tangent kernel)} \tag{24}$$

$$= \eta \, \boldsymbol{\Theta}_t([\mathcal{X}, \mathcal{X}'], [\mathcal{X}, \mathcal{X}']) \nabla_{[\mathcal{F}_t, \mathcal{F}'_t]} \frac{1}{2}\|\mathcal{F}_t - \mathcal{Y}\|_F^2 \qquad \text{(Loss function)} \tag{25}$$

$$= -\eta \, \boldsymbol{\Theta}_t([\mathcal{X}, \mathcal{X}'], [\mathcal{X}, \mathcal{X}']) [\mathcal{R}_t, \mathbf{0}] \qquad \text{(Compute gradient)} \tag{26}$$

where

$$\boldsymbol{\Theta}_t([\mathcal{X}, \mathcal{X}'], [\mathcal{X}, \mathcal{X}']) \triangleq \begin{bmatrix} \boldsymbol{\Theta}_t(\mathcal{X}, \mathcal{X}) & \boldsymbol{\Theta}_t(\mathcal{X}, \mathcal{X}') \\ \boldsymbol{\Theta}_t(\mathcal{X}', \mathcal{X}) & \boldsymbol{\Theta}_t(\mathcal{X}', \mathcal{X}') \end{bmatrix} \tag{27}$$

is the NTK matrix in $\mathbb{R}^{n \times n}$. Discretizing the residual dynamics with step size $\Delta t = 1$ and rearranging the equation gives

$$[\mathcal{R}_{t+1}, \mathcal{R}'_{t+1}] = -\eta \begin{bmatrix} \boldsymbol{\Theta}_t(\mathcal{X}, \mathcal{X}) & \boldsymbol{\Theta}_t(\mathcal{X}, \mathcal{X}') \\ \boldsymbol{\Theta}_t(\mathcal{X}', \mathcal{X}) & \boldsymbol{\Theta}_t(\mathcal{X}', \mathcal{X}') \end{bmatrix} [\mathcal{R}_t, \mathbf{0}] + [\mathcal{R}_t, \mathcal{R}'_t]. \tag{28}$$

For the training set, it can be written as

$$\mathcal{R}_{t+1} = \mathcal{R}_t - \eta \boldsymbol{\Theta}_t(\mathcal{X}, \mathcal{X}) \mathcal{R}_t \tag{29}$$

which has been widely used for analyzing convergence of empirical risk in optimization. For the testing set, it can be written as

$$\mathcal{R}'_{t+1} = \mathcal{R}'_t - \eta \boldsymbol{\Theta}_t(\mathcal{X}', \mathcal{X}) \mathcal{R}_t \tag{30}$$

which is of our interest since it characterizes the learn function on arbitrary unseen samples and reveals how GNNs generalize.

## B.2 PROOF OF THEOREM 2: CONVERGENCE OF RP, CONNECTION WITH KERNEL REGRESSION

In this proof we examine generic RP iterations of the form given by

$$\left[\mathcal{R}_{t+1}, \mathcal{R}'_{t+1}\right] = -\eta \mathbf{S}[\mathcal{R}_t, \mathbf{0}] + [\mathcal{R}_t, \mathcal{R}'_t], \tag{31}$$

where $\mathbf{S}$ is an arbitrary symmetric matrix as similarity measure ($\mathbf{S} = \mathbf{A}^K$ for the basic version of RP in (6)) with block structure aligned with the dimensions of $\mathcal{R}_t$ and $\mathcal{R}'_t$ respectively

$$\mathbf{S} = \left[ \begin{array}{cc} \mathbf{S}_{\mathcal{X}\mathcal{X}} & \mathbf{S}_{\mathcal{X}\mathcal{X}'} \\ \mathbf{S}_{\mathcal{X}'\mathcal{X}} & \mathbf{S}_{\mathcal{X}'\mathcal{X}'} \end{array} \right], \tag{32}$$

where $\mathbf{S}_{\mathcal{X}\mathcal{X}} \in \mathbb{R}^{n_l \times n_l}$ is a principal submatrix of $\mathbf{S} \in \mathbb{R}^{n \times n}$ corresponding to the training set. According to (31), the explicit form of training residuals can be written as

$$\begin{aligned} \mathcal{R}_{t+1} &= (\mathbf{I}_{n_l} - \eta \mathbf{S}_{\mathcal{X}\mathcal{X}}) \mathcal{R}_t \\ &= (\mathbf{I}_{n_l} - \eta \mathbf{S}_{\mathcal{X}\mathcal{X}})^{t+1} \mathcal{Y} \end{aligned} \tag{33}$$

and testing residuals can be written as

$$\begin{aligned} \mathcal{R}'_{t+1} &= -\eta \mathbf{S}_{\mathcal{X}'\mathcal{X}} \mathcal{R}_t + \mathcal{R}'_t = -\eta \sum_{i=0}^{t} \mathbf{S}_{\mathcal{X}'\mathcal{X}} \mathcal{R}_i \\ &= -\eta \mathbf{S}_{\mathcal{X}'\mathcal{X}} \sum_{i=0}^{t} (\mathbf{I}_{n_l} - \eta \mathbf{S}_{\mathcal{X}\mathcal{X}})^i \mathcal{Y} \end{aligned} \tag{34}$$

To analyze their convergence, we consider three variants based on the property of $\mathbf{S}_{\mathcal{X}\mathcal{X}}$: 1) $\mathbf{S}_{\mathcal{X}\mathcal{X}}$ is positive definite; 2) $\mathbf{S}_{\mathcal{X}\mathcal{X}}$ is positive semi-definite (but not positive definite); 3) $\mathbf{S}_{\mathcal{X}\mathcal{X}}$ is not positive semi-definite.

For the first variant, we stipulate $\mathbf{S}_{\mathcal{X}\mathcal{X}}$ is positive definite. In this case, sufficiently small step size $\eta < 2/\sigma_{max}[\mathbf{S}_{\mathcal{X}\mathcal{X}}]$, where $\sigma_{max}[\mathbf{S}_{\mathcal{X}\mathcal{X}}]$ is the largest eigenvalue of $\mathbf{S}_{\mathcal{X}\mathcal{X}}$, ensures each diagonal element in $\mathbf{I}_{n_l} - \eta \mathbf{S}_{\mathcal{X}\mathcal{X}}$ to lie between $(-1, 1)$. Therefore, as $t \to \infty$, the power and geometric series of $\mathbf{I}_{n_l} - \eta \mathbf{S}_{\mathcal{X}\mathcal{X}}$ converge to

$$\begin{aligned} (\mathbf{I}_{n_l} - \eta \mathbf{S}_{\mathcal{X}\mathcal{X}})^{t+1} &\to \mathbf{Q}(\mathbf{I}_{n_l} - \eta \mathbf{\Lambda}[\mathbf{S}_{\mathcal{X}\mathcal{X}}])^{\infty} \mathbf{Q}^{-1} = \mathbf{0}, \\ \sum_{i=0}^{t} (\mathbf{I}_{n_l} - \eta \mathbf{S}_{\mathcal{X}\mathcal{X}})^i &\to (\mathbf{I}_{n_l} - (\mathbf{I}_{n_l} - \eta \mathbf{S}_{\mathcal{X}\mathcal{X}}))^{-1} = (\eta \mathbf{S}_{\mathcal{X}\mathcal{X}})^{-1}. \end{aligned} \tag{35}$$

Plugging them back into (33) and (34) gives us

$$\mathcal{R}_t \to \mathbf{0}, \quad \mathcal{R}'_t \to -\mathbf{S}_{\mathcal{X}'\mathcal{X}} \mathbf{S}_{\mathcal{X}\mathcal{X}}^{-1} \mathcal{Y}. \tag{36}$$

Correspondingly, by noting that $\mathcal{Y}' = \mathbf{0}$ at initialization, we have

$$\mathcal{F}_t \to \mathcal{Y}, \quad \mathcal{F}'_t \to \mathbf{S}_{\mathcal{X}'\mathcal{X}} \mathbf{S}_{\mathcal{X}\mathcal{X}}^{-1} \mathcal{Y}. \tag{37}$$

Namely, for positive definite $\mathbf{S}_{\mathcal{X}\mathcal{X}}$, the converged model predictions $\mathcal{F}_{\infty}$ perfectly fit the training labels $\mathcal{Y}$, and we further assume that $\mathbf{S}$ is positive definite, equation 37 is equivalent to the solution of kernel regression with respect to the kernel $\kappa(\boldsymbol{x}, \boldsymbol{x}') = \mathbf{A}_{\boldsymbol{x}\boldsymbol{x}'}^K$.

For the second variant, we stipulate that $\mathbf{S}_{\mathcal{X}\mathcal{X}}$ is positive semi-definite (but not positive definite), which implies that $\mathbf{S}_{\mathcal{X}\mathcal{X}} = \mathbf{B}\mathbf{B}^{\top}$ for some tall matrix $\mathbf{B} \in \mathbb{R}^{n_l \times r}$ that is full column rank. Suppose $\mathbf{P}_{C(\mathbf{B})} = \mathbf{B}(\mathbf{B}^{\top}\mathbf{B})^{-1}\mathbf{B}^{\top}$ and $\mathbf{P}_{N(\mathbf{B}^{\top})} = \mathbf{I}_{n_l} - \mathbf{P}_{C(\mathbf{B})}$ denotes the projection matrices onto the column space of $\mathbf{B}$ and null space of $\mathbf{B}^{\top}$. By their construction,

$$\mathbf{P}_{C(\mathbf{B})}\mathbf{B} = \mathbf{B}, \quad \mathbf{P}_{N(\mathbf{B}^{\top})}\mathbf{B} = \mathbf{0}. \tag{38}$$

For training residuals, we may form the decomposition

$$\begin{aligned} \mathcal{R}_{t+1} &= (\mathbf{I}_{n_l} - \eta \mathbf{B}\mathbf{B}^{\top})^{t+1}\mathcal{Y} \\ &= (\mathbf{I}_{n_l} - \eta \mathbf{B}\mathbf{B}^{\top})^{t+1}\mathbf{P}_{C(\mathbf{B})}\mathcal{Y} + (\mathbf{I}_{n_l} - \eta \mathbf{B}\mathbf{B}^{\top})^{t+1}\mathbf{P}_{N(\mathbf{B}^{\top})}\mathcal{Y} \end{aligned} \tag{39}$$

Table 2: Summary of the convergence of residual propagation. The convergence of model predictions can be inferred by $\mathcal{F}_t = \mathcal{Y} - \mathcal{R}_t$ and $\mathcal{F}'_t = -\mathcal{R}'_t$.

| $\mathbf{S}_{\mathcal{XX}}$ | $\mathbf{S}$ | Convergence of $\mathcal{R}_t$ | Convergence of $\mathcal{R}'_t$ | Counterpart |
|---|---|---|---|---|
| PD | PSD | $\mathbf{0}$ | $-\mathbf{S}_{\mathcal{X'X}}\mathbf{S}_{\mathcal{XX}}^{-1}\mathcal{Y}$ | Kernel Regression |
| PD | not PSD | $\mathbf{0}$ | $-\mathbf{S}_{\mathcal{X'X}}\mathbf{S}_{\mathcal{XX}}^{-1}\mathcal{Y}$ | Unique |
| PSD (but not PD) | PSD | $\left(\mathbf{I}_{n_l} - \mathbf{B}(\mathbf{B}^\top\mathbf{B})^{-1}\mathbf{B}^\top\right)\mathcal{Y}$ | $-\mathbf{B}'(\mathbf{B}^\top\mathbf{B})^{-1}\mathbf{B}^\top\mathcal{Y}$ | Linear Regression |
| PSD (but not PD) | not PSD | $\left(\mathbf{I}_{n_l} - \mathbf{B}(\mathbf{B}^\top\mathbf{B})^{-1}\mathbf{B}^\top\right)\mathcal{Y}$ | Not Converge | Unique |
| not PSD | not PSD | Not Converge | Not Converge | Unique |

It follows that, for the first term on the RHS of (39),

$$
\begin{aligned}
(\mathbf{I}_{n_l} - \eta\mathbf{B}\mathbf{B}^\top)^{t+1}\mathbf{P}_{C(\mathbf{B})}\mathcal{Y} &= (\mathbf{I}_{n_l} - \eta\mathbf{B}\mathbf{B}^\top)^{t+1}\mathbf{B}\mathbf{B}^\dagger\mathcal{Y} \\
&= (\mathbf{I}_{n_l} - \eta\mathbf{B}\mathbf{B}^\top)^t(\mathbf{I}_{n_l} - \eta\mathbf{B}\mathbf{B}^\top)\mathbf{B}\mathbf{B}^\dagger\mathcal{Y} \\
&= (\mathbf{I}_{n_l} - \eta\mathbf{B}\mathbf{B}^\top)^t\mathbf{B}(\mathbf{I}_r - \eta\mathbf{B}^\top\mathbf{B})\mathbf{B}^\dagger\mathcal{Y} \\
&= \mathbf{B}(\mathbf{I}_r - \eta\mathbf{B}^\top\mathbf{B})^{t+1}\mathbf{B}^\dagger\mathcal{Y} \\
&\to \mathbf{0}, \quad (40)
\end{aligned}
$$

where $\mathbf{B}^\dagger = (\mathbf{B}^\top\mathbf{B})^{-1}\mathbf{B}^\top$ is the pseudo inverse of $\mathbf{B}$. The convergence also requires $\eta < 2/\sigma_{max}[\mathbf{S}_{\mathcal{XX}}]$.

For the second term on the RHS of (39), we have

$$
\begin{aligned}
(\mathbf{I} - \eta\mathbf{B}\mathbf{B}^\top)^{t+1}\mathbf{P}_{N(\mathbf{B}^\top)}\mathcal{Y} &= (\mathbf{I} - \eta\mathbf{B}\mathbf{B}^\top)^t(\mathbf{P}_{N(\mathbf{B}^\top)} - \eta\mathbf{B}(\mathbf{P}_{N(\mathbf{B}^\top)}\mathbf{B})^\top)\mathcal{Y} \\
&= (\mathbf{I} - \eta\mathbf{B}\mathbf{B}^\top)^t\mathbf{P}_{N(\mathbf{B}^\top)}\mathcal{Y} \\
&= \mathbf{P}_{N(\mathbf{B}^\top)}\mathcal{Y} \quad (41)
\end{aligned}
$$

It follows that

$$
\mathcal{R}_t \to \mathbf{P}_{N(\mathbf{B}^\top)}\mathcal{Y} = \left(\mathbf{I}_{n_l} - \mathbf{B}(\mathbf{B}^\top\mathbf{B})^{-1}\mathbf{B}^\top\right)\mathcal{Y}, \quad (42)
$$

which is equivalent to optimal training residuals of linear regression (Boyd & Vandenberghe, 2004).

However, in our case, the testing residuals will not necessarily converge for arbitrary $\mathbf{S}_{\mathcal{X'X}}$ since the geometric series in (34) diverges outside $(-1, 1)$. Nevertheless, we can further stipulate $\mathbf{S}$ is also positive semi-definite. In this case, we have $\mathbf{S}_{\mathcal{X'X}} = \mathbf{B}'\mathbf{B}^\top$ for another tall matrix $\mathbf{B}' \in \mathbb{R}^{n_u \times r}$. Correspondingly, the testing residuals can be written as

$$
\begin{aligned}
\mathcal{R}'_{t+1} &= -\eta\mathbf{B}'\mathbf{B}^\top\sum_{i=0}^t \left(\mathbf{I}_{n_l} - \eta\mathbf{B}\mathbf{B}^\top\right)^i\mathcal{Y} \\
&= -\eta\mathbf{B}'\sum_{i=0}^t \left(\mathbf{I}_r - \eta\mathbf{B}^\top\mathbf{B}\right)^i\mathbf{B}^\top\mathcal{Y} \\
&= -\eta\mathbf{B}'\sum_{i=0}^t \left(\mathbf{I}_r - \eta\mathbf{B}^\top\mathbf{B}\right)^i\mathbf{B}^\top\mathcal{Y} \\
&\to -\eta\mathbf{B}'(\mathbf{I}_r - (\mathbf{I}_r - \eta\mathbf{B}^\top\mathbf{B}))^{-1}\mathbf{B}^\top\mathcal{Y} \\
&= -\mathbf{B}'(\mathbf{B}^\top\mathbf{B})^{-1}\mathbf{B}^\top\mathcal{Y}. \quad (43)
\end{aligned}
$$

For the last variant, if $\mathbf{S}_{\mathcal{XX}}$ is not positive semi-definite, it is no longer possible to guarantee convergence from arbitrary initializations. Rather we can only establish that solutions of the form described above can serve as fixed points of the iterations.

To summarize, for the RP algorithm in (6) where $\mathbf{S} = \mathbf{A}^K$, if step size is sufficiently small $\eta < 2\sigma_{max}^{-1}[\mathbf{A}_{\mathcal{XX}}^K]$, both $\mathcal{R}_t$ and $\mathcal{R}'_t$ converge as $t \to \infty$ for positive definite $\mathbf{A}_{\mathcal{XX}}^K$ or positive semi-definite $\mathbf{A}^K$. In practice, one can choose $K$ as an even number or letting $\mathbf{A} \leftarrow \alpha\mathbf{A} + (1-\alpha)\mathbf{I}_n$ for $\alpha >= \frac{1}{2}$ to enforce positive semi-definiteness of $\mathbf{A}^K$. However, in practice, we did not find that test performance was compromised when the algorithm does not converge. Table 2 gives a clear overview of the convergence of RP and connection with existing learning algorithms.

### B.3 COMPUTATION OF NODE-LEVEL GNTK

We present the recurrent formula for computing of GNTK in node-level tasks. Suppose the GNN is denoted as $f(\boldsymbol{x}_i; \mathbf{A}) \in \mathbb{R}$ where the input is node feature $\boldsymbol{x}_i$, the graph structure $\mathbf{A}$ is used for cross-instance feature propagation at each layer, weights are $\mathbf{W}$. The GNTK in node-level tasks is defined as

$$\boldsymbol{\Theta}_t(\boldsymbol{x}_i, \boldsymbol{x}_j; \mathbf{A}) = \left\langle \frac{\partial f_t(\boldsymbol{x}_i; \mathbf{A})}{\partial \mathbf{W}_t}, \frac{\partial f_t(\boldsymbol{x}_j; \mathbf{A})}{\partial \mathbf{W}_t} \right\rangle, \tag{44}$$

for a pair of nodes (i.e. data points) at optimization time index $t$. Intuitively, the kernel function measures quantifies similarity between seen and unseen instances based on how differently their outputs given by the GNN change by an infinitesimal perturbation of weights (which we will show is biased by the adjacency matrix used for feature propagation in GNNs).

Note that we consider transductive learning here, which is more convenient. For inductive learning, one should replace $\bar{\mathcal{X}}$ with $\mathcal{X}$, and $\mathbf{A}$ with its submatrix $\mathbf{A}_{\mathcal{X}\mathcal{X}}$. Let us denote GNTK with/without feature propagation at $\ell$-th layer as $\boldsymbol{\Theta}^{(\ell)}/\bar{\boldsymbol{\Theta}}^{(\ell)}$, covariance matrix of outputs of $\ell$-th layer with/without feature propagation as $\boldsymbol{\Sigma}^{(\ell)}/\bar{\boldsymbol{\Sigma}}^{(\ell)}$, covariance matrix of the derivative to $\ell$-th layer layer as $\dot{\boldsymbol{\Sigma}}^{(\ell)}$. The recurrent formula for computing node-level GNTK for infinitely-wide GNNs can be written as the following. The initialization of GNTK for graph regression is given as

$$\bar{\boldsymbol{\Theta}}^{(1)}(\boldsymbol{x}_i, \boldsymbol{x}_j; \mathbf{A}) = \bar{\boldsymbol{\Sigma}}^{(1)}(\boldsymbol{x}_i, \boldsymbol{x}_j; \mathbf{A}) = \boldsymbol{x}_i^\top \boldsymbol{x}_j, \tag{45}$$

We also write matrix form of computing GNTK here in order to give clearer intuitions of how the adjacency matrix can be naturally encoded into the computation of $\boldsymbol{\Theta}$ in node regression tasks:

$$\bar{\boldsymbol{\Theta}}^{(1)}(\bar{\mathcal{X}}, \bar{\mathcal{X}}; \mathbf{A}) = \bar{\boldsymbol{\Sigma}}^{(1)}(\bar{\mathcal{X}}, \bar{\mathcal{X}}; \mathbf{A}) = \bar{\mathcal{X}}^\top \bar{\mathcal{X}}. \tag{46}$$

**Feature propagation.** The feature propagation operation (i.e. $\mathbf{Z} \leftarrow \mathbf{A}\mathbf{Z}$) at each layer corresponds to

$$\begin{aligned}
\boldsymbol{\Sigma}^{(\ell-1)}(\boldsymbol{x}_i, \boldsymbol{x}_j; \mathbf{A}) &= \sum_{i' \in \mathcal{N}_i} \sum_{j' \in \mathcal{N}_j} \mathbf{A}_{ii'} \mathbf{A}_{jj'} \bar{\boldsymbol{\Sigma}}^{(\ell-1)}(\boldsymbol{x}_{i'}, \boldsymbol{x}_{j'}; \mathbf{A}) \\
\boldsymbol{\Theta}^{(\ell-1)}(\boldsymbol{x}_i, \boldsymbol{x}_j; \mathbf{A}) &= \sum_{i' \in \mathcal{N}_i} \sum_{j' \in \mathcal{N}_j} \mathbf{A}_{ii'} \mathbf{A}_{jj'} \bar{\boldsymbol{\Theta}}^{(\ell-1)}(\boldsymbol{x}_{i'}, \boldsymbol{x}_{j'}; \mathbf{A}),
\end{aligned} \tag{47}$$

where $\mathcal{N}_i$ denote neighboring nodes of $\boldsymbol{x}_i$ including $\boldsymbol{x}_i$ itself. For GCN where $\mathbf{A}$ is defined as a symmetric normalized adjacency matrix, we have $\mathbf{A}_{ij} = \mathbf{A}_{ji} = 1/\sqrt{d_i d_j}$. The above equation can be compactly described in a matrix form:

$$\begin{aligned}
\boldsymbol{\Sigma}^{(\ell-1)}(\bar{\mathcal{X}}, \bar{\mathcal{X}}; \mathbf{A}) &= \mathbf{A}\, \bar{\boldsymbol{\Sigma}}^{(\ell-1)}(\bar{\mathcal{X}}, \bar{\mathcal{X}}; \mathbf{A})\, \mathbf{A} \\
\boldsymbol{\Theta}^{(\ell-1)}(\bar{\mathcal{X}}, \bar{\mathcal{X}}; \mathbf{A}) &= \mathbf{A}\, \bar{\boldsymbol{\Theta}}^{(\ell-1)}(\bar{\mathcal{X}}, \bar{\mathcal{X}}; \mathbf{A})\, \mathbf{A}.
\end{aligned} \tag{48}$$

The multiplication of $\mathbf{A}$ before $\bar{\boldsymbol{\Theta}}^{(\ell-1)}(\bar{\mathcal{X}}, \bar{\mathcal{X}}; \mathbf{A})$ gives row-wise weighted summation and after $\bar{\boldsymbol{\Theta}}^{(\ell-1)}(\bar{\mathcal{X}}, \bar{\mathcal{X}}; \mathbf{A})$ gives column-wise weighted summation.

**Feature transformation.** Let $\mathcal{T}$ and $\dot{\mathcal{T}}$ be functions from $2 \times 2$ positive semi-definite matrices $\boldsymbol{\Lambda}$ to $\mathbb{R}$ given by

$$\begin{cases} \mathcal{T}(\boldsymbol{\Lambda}) = \mathbb{E}[\sigma(a)\sigma(b)] \\ \dot{\mathcal{T}}(\boldsymbol{\Lambda}) = \mathbb{E}[\sigma'(a)\sigma'(b)] \end{cases} \quad (a, b) \sim \mathcal{N}(0, \boldsymbol{\Lambda}) \tag{49}$$

where $\sigma'$ is the derivative of the activation function. Then, the feature transformation process (i.e. $\mathbf{Z} = \sigma(\mathbf{Z}\mathbf{W})$) then corresponds to:

$$\begin{aligned}
\bar{\boldsymbol{\Sigma}}^{(\ell)}(\boldsymbol{x}_i, \boldsymbol{x}_j; \mathbf{A}) &= c_\sigma \mathcal{T}\left( \begin{matrix} \boldsymbol{\Sigma}^{(\ell-1)}(\boldsymbol{x}_i, \boldsymbol{x}_i; \mathbf{A}) & \boldsymbol{\Sigma}^{(\ell-1)}(\boldsymbol{x}_i, \boldsymbol{x}_j; \mathbf{A}) \\ \boldsymbol{\Sigma}^{(\ell-1)}(\boldsymbol{x}_j, \boldsymbol{x}_i; \mathbf{A}) & \boldsymbol{\Sigma}^{(\ell-1)}(\boldsymbol{x}_j, \boldsymbol{x}_j; \mathbf{A}) \end{matrix} \right), \\
\dot{\boldsymbol{\Sigma}}^{(\ell)}(\boldsymbol{x}_i, \boldsymbol{x}_j; \mathbf{A}) &= c_\sigma \dot{\mathcal{T}}\left( \begin{matrix} \boldsymbol{\Sigma}^{(\ell-1)}(\boldsymbol{x}_i, \boldsymbol{x}_i; \mathbf{A}) & \boldsymbol{\Sigma}^{(\ell-1)}(\boldsymbol{x}_i, \boldsymbol{x}_j; \mathbf{A}) \\ \boldsymbol{\Sigma}^{(\ell-1)}(\boldsymbol{x}_j, \boldsymbol{x}_i; \mathbf{A}) & \boldsymbol{\Sigma}^{(\ell-1)}(\boldsymbol{x}_j, \boldsymbol{x}_j; \mathbf{A}) \end{matrix} \right).
\end{aligned} \tag{50}$$

The layer-wise computation for $\boldsymbol{\Theta}$ is given as

$$\bar{\boldsymbol{\Theta}}^{(\ell)}\left(\bar{\mathcal{X}}, \bar{\mathcal{X}}; \mathbf{A}\right) = \boldsymbol{\Theta}^{(\ell-1)}\left(\bar{\mathcal{X}}, \bar{\mathcal{X}}; \mathbf{A}\right) \odot \dot{\boldsymbol{\Sigma}}^{(\ell)}\left(\bar{\mathcal{X}}, \bar{\mathcal{X}}; \mathbf{A}\right) + \bar{\boldsymbol{\Sigma}}^{(\ell)}\left(\bar{\mathcal{X}}, \bar{\mathcal{X}}; \mathbf{A}\right), \tag{51}$$

where $\odot$ denotes Hadamard product. For a $\ell$-layer GNN, the corresponding GNTK is given by $\boldsymbol{\Theta}^{(\ell)}(\bar{\mathcal{X}}, \bar{\mathcal{X}}; \mathbf{A})$, which is abbreviated as $\boldsymbol{\Theta}^{(\ell)}$ in the main text.

### B.4 PROOF OF THEOREM 5: TWO-LAYER GNN

Consider a two-layer GNN defined as

$$\{f(\boldsymbol{x}; \mathbf{A})\}_{\boldsymbol{x} \in \bar{\mathcal{X}}} = [\mathcal{F}, \mathcal{F}'] = \mathbf{A} \frac{1}{\sqrt{m}} \sigma\left(\mathbf{A}\bar{\mathcal{X}}\mathbf{W}^{(1)}\right) \mathbf{W}^{(2)} \tag{52}$$

where $m$ is the width and $\sigma$ is ReLU activation. Following previous works that analyze two-layer fully-connected neural networks (Arora et al., 2019; Du et al., 2019c), we consider optimization of the first layer weights $\mathbf{W}^{(1)}$ while fixing the second-layer weights. In this case, the two-layer GNTK is defined as

$$\boldsymbol{\Theta}^{(2)}(\boldsymbol{x}_i, \boldsymbol{x}_j; \mathbf{A}) = \left\langle \frac{\partial f(\boldsymbol{x}_i; \mathbf{A}, \mathbf{W})}{\partial \mathbf{W}^{(1)}}, \frac{\partial f(\boldsymbol{x}_j; \mathbf{A}, \mathbf{W})}{\partial \mathbf{W}^{(1)}} \right\rangle. \tag{53}$$

We can derive the the explicit form formula for computing it in overparameterized regime based on the general formula for arbitrarily deep GNNs given in Appendix. B.3 (or directly calculating $\partial f(\boldsymbol{x}_i; \mathbf{A}, \mathbf{W})/\partial \mathbf{W}^{(1)}$ by using chain rule). Consequently, we have

$$\boldsymbol{\Theta}^{(2)}(\boldsymbol{x}_i, \boldsymbol{x}_j; \mathbf{A}) = \sum_{i' \in \mathcal{N}(i)} \sum_{j' \in \mathcal{N}(j)} \mathbf{A}_{ii'} \mathbf{A}_{jj'} \left([\mathbf{A}\bar{\mathcal{X}}]_{i'}^\top [\mathbf{A}\bar{\mathcal{X}}]_{j'}\right)$$
$$\mathbb{E}_{\boldsymbol{w} \sim \mathcal{N}(0,1)} \left[\mathbb{1}\left\{\boldsymbol{w}^\top [\mathbf{A}\bar{\mathcal{X}}]_{i'} \geq 0, \boldsymbol{w}^\top [\mathbf{A}\bar{\mathcal{X}}]_{j'} \geq 0\right\}\right] \tag{54}$$

where $\mathcal{N}(i)$ denote the set of neighboring nodes, $\boldsymbol{w} \in \mathbb{R}^d$ is sampled from Gaussian distribution, and $\mathbb{1}$ is indicator function.

**Case 1.** If we stipulate the input features are represented by an identity matrix, i.e. $\bar{\mathcal{X}} = \mathbf{I}_n$, it follows that

$$\boldsymbol{\Theta}^{(2)}(\boldsymbol{x}_i, \boldsymbol{x}_j; \mathbf{A}) = \sum_{i' \in \mathcal{N}(i)} \sum_{j' \in \mathcal{N}(j)} \mathbf{A}_{ii'} \mathbf{A}_{jj'} \left(\mathbf{A}_{i'}^\top \mathbf{A}_{j'}\right) \mathbb{E}_{\boldsymbol{w} \sim \mathcal{N}(0,1)} \left[\mathbb{1}\left\{\boldsymbol{w}^\top \mathbf{A}_{i'} \geq 0, \boldsymbol{w}^\top \mathbf{A}_{j'} \geq 0\right\}\right]$$
$$= \sum_{i' \in \mathcal{N}(i)} \sum_{j' \in \mathcal{N}(j)} \mathbf{A}_{ii'} \mathbf{A}_{jj'} \frac{\mathbf{A}_{i'}^\top \mathbf{A}_{j'}(\pi - \arccos(\frac{\mathbf{A}_{i'}^\top \mathbf{A}_{j'}}{\|\mathbf{A}_{i'}\|\|\mathbf{A}_{j'}\|}))}{2\pi}. \tag{55}$$

The above equation can be written neatly in the matrix form

$$\boldsymbol{\Theta}^{(2)}(\bar{\mathcal{X}}, \bar{\mathcal{X}}; \mathbf{A}) = \mathbf{A}(\mathbf{A}^2 \odot \mathbf{S})\mathbf{A}, \quad \text{where} \quad \mathbf{S}_{ij} = (\pi - \arccos(\frac{\mathbf{A}_i^\top \mathbf{A}_j}{\|\mathbf{A}_i\|\|\mathbf{A}_j\|}))/2\pi. \tag{56}$$

The matrix $\mathbf{S}$ reweights each entry in $\mathbf{A}^2$ by the similarity of neighborhood patterns of $\boldsymbol{x}_i$ and $\boldsymbol{x}_j$. Substituting it to (12) gives us the learning dynamics of overparameterized two-layer GNN

$$\left[\mathcal{R}_{t+1}, \mathcal{R}'_{t+1}\right] = -\eta \mathbf{A}(\mathbf{A}^2 \odot \mathbf{S})\mathbf{A}[\mathcal{R}_t, \mathbf{0}] + [\mathcal{R}_t, \mathcal{R}'_t]. \tag{57}$$

**Case 2.** If we stipulate the input features are node embeddings from spectral decomposition of a full-rank adjacency matrix, i.e. $\bar{\mathcal{X}} \triangleq \arg\min_{\mathbf{B}} \|\mathbf{A} - \mathbf{B}\mathbf{B}^\top\|_F^2$. We have

$$\boldsymbol{\Theta}^{(2)}(\boldsymbol{x}_i, \boldsymbol{x}_j; \mathbf{A})$$
$$= \sum_{i' \in \mathcal{N}(i)} \sum_{j' \in \mathcal{N}(j)} \mathbf{A}_{ii'} \mathbf{A}_{jj'} \mathbf{A}_{i'j'}^3 \mathbb{E}_{\boldsymbol{w} \sim \mathcal{N}(0,1)} \left[\mathbb{1}\left\{\boldsymbol{w}^\top [\mathbf{A}\bar{\mathcal{X}}]_{i'} \geq 0, \boldsymbol{w}^\top [\mathbf{A}\bar{\mathcal{X}}]_{j'} \geq 0\right\}\right]$$
$$= \sum_{i' \in \mathcal{N}(i)} \sum_{j' \in \mathcal{N}(j)} \mathbf{A}_{ii'} \mathbf{A}_{jj'} \frac{\mathbf{A}_{i'j'}^3(\pi - \arccos(\frac{[\mathbf{A}\bar{\mathcal{X}}]_{i'}^\top [\mathbf{A}\bar{\mathcal{X}}]_{j'}}{\|[\mathbf{A}\bar{\mathcal{X}}]_{i'}\|\|[\mathbf{A}\bar{\mathcal{X}}]_{j'}\|}))}{2\pi}. \tag{58}$$

The above equation can be written neatly in the matrix form

$$\mathbf{\Theta}^{(2)}(\bar{\mathcal{X}}, \bar{\mathcal{X}}; \mathbf{A}) = \mathbf{A}(\mathbf{A}^3 \odot \tilde{\mathbf{S}})\mathbf{A}, \quad \text{where} \quad \tilde{\mathbf{S}}_{ij} = (\pi - \arccos(\frac{[\mathbf{A}\bar{\mathcal{X}}]_i^\top [\mathbf{A}\bar{\mathcal{X}}]_j}{\|[\mathbf{A}\bar{\mathcal{X}}]_i\|\|[\mathbf{A}\bar{\mathcal{X}}]_j\|}))/2\pi. \quad (59)$$

The matrix $\tilde{\mathbf{S}}$ reweights each entry in $\mathbf{A}^3$ by the similarity of aggregated node embeddings $[\mathbf{A}\bar{\mathcal{X}}]_i$ and $[\mathbf{A}\bar{\mathcal{X}}]_j$. Substituting it to (12) gives us the learning dynamics of overparameterized two-layer GNN

$$\left[\mathcal{R}_{t+1}, \mathcal{R}'_{t+1}\right] = -\eta\mathbf{A}(\mathbf{A}^3 \odot \tilde{\mathbf{S}})\mathbf{A}[\mathcal{R}_t, \mathbf{0}] + [\mathcal{R}_t, \mathcal{R}'_t]. \quad (60)$$

### B.5 PROOF OF THEOREM 6: ARBITRARILY DEEP GNN

Before analyzing the learning dynamics of deep GNNs, let us first consider an arbitrarily deep standard fully-connected neural networks, which is denoted as $\mathrm{MLP}(\bar{\mathcal{X}})$ with weights $\mathbf{W}$. We also denote its corresponding NTK in overparameterized regime as $\mathbf{\Theta}^{(\ell)}(\bar{\mathcal{X}}, \bar{\mathcal{X}})$. For input features as an identity matrix $\bar{\mathcal{X}} = \mathbf{I}_n$, we have the following permutation invariant property of $\mathbf{\Theta}^{(\ell)}(\bar{\mathcal{X}}, \bar{\mathcal{X}})$.

**Lemma 10** (Permutation Invariance of NTK). *For arbitrarily deep and infinitely wide fully-connected neural networks* $\mathrm{MLP}(\bar{\mathcal{X}})$ *with ReLU activation and standard NTK parameterization, the corresponding NTK* $\mathbf{\Theta}^{(\ell)}(\bar{\mathcal{X}}, \bar{\mathcal{X}})$ *for onehot vector inputs is permutation invariant. Namely for arbitrary permutation function* $\psi$ *on input set* $\bar{\mathcal{X}}$

$$\mathbf{\Theta}^{(\ell)}(\psi(\bar{\mathcal{X}}), \psi(\bar{\mathcal{X}})) = \mathbf{\Theta}^{(\ell)}(\bar{\mathcal{X}}, \bar{\mathcal{X}}). \quad (61)$$

*Proof.* Recall that in the explicit form computation for NTK of fully-connected neural networks (Jacot et al., 2018; Lee et al., 2019), the NTK and NNGP for a $(\ell + 1)$-layer network is produced from the NTK and NNGP for a $\ell$-layer network. Therefore, to prove the permutation invariance of NTK of an arbitrarily deep fully-connected neural network, we need only to prove the permutation invariance of $\mathbf{\Theta}^{(1)}$ and $\mathbf{\Sigma}^{(1)}$, which are defined as the inner product of input features

$$\mathbf{\Theta}^{(1)}(\bar{\mathcal{X}}, \bar{\mathcal{X}}) = \mathbf{\Sigma}^{(1)}(\bar{\mathcal{X}}, \bar{\mathcal{X}}) = \bar{\mathcal{X}}\bar{\mathcal{X}}^\top. \quad (62)$$

Since arbitrary permutation function $\psi$ on $\bar{\mathcal{X}}$ can be expressed as

$$\psi(\bar{\mathcal{X}}) = \prod_{i=0}^{c} \mathbf{T}_i\bar{\mathcal{X}} \quad (63)$$

for some row-interchanging elementary matrices $\{\mathbf{T}_i\}_{i=0}^{c}$. It follows that

$$
\begin{aligned}
\mathbf{\Theta}^{(1)}(\psi(\bar{\mathcal{X}}), \psi(\bar{\mathcal{X}})) &= \prod_{i=0}^{c} \mathbf{T}_i\bar{\mathcal{X}}(\prod_{i=0}^{c} \mathbf{T}_i\bar{\mathcal{X}})^\top \\
&= \prod_{i=0}^{c} \mathbf{T}_i\bar{\mathcal{X}}\bar{\mathcal{X}}^\top \prod_{i=0}^{c} \mathbf{T}_{c-i}^\top \\
&= \mathbf{T}_0 \cdots \mathbf{T}_c\mathbf{T}_c^\top \cdots \mathbf{T}_0^\top \\
&= \mathbf{I}_n \\
&= \mathbf{\Theta}^{(1)}(\bar{\mathcal{X}}, \bar{\mathcal{X}}).
\end{aligned} \quad (64)
$$

By induction, it follows that $\mathbf{\Theta}^{(\ell)}(\bar{\mathcal{X}}, \bar{\mathcal{X}})$ is also permutation invariant. □

Now, let us consider a deep GNN defined as

$$[\mathcal{F}, \mathcal{F}'] = \mathbf{A}^\ell \mathrm{MLP}(\bar{\mathcal{X}}) \quad (65)$$

where $\mathrm{MLP}(\bar{\mathcal{X}})$ is a $L$-layer infinitely-wide fully-connected neural network with ReLU activation. Then, the corresponding node-level GNTK matrix is defined as

$$\left\langle \frac{\partial \mathbf{A}^\ell \mathrm{MLP}(\bar{\mathcal{X}})}{\partial \mathbf{W}}, \frac{\partial \mathbf{A}^\ell \mathrm{MLP}(\bar{\mathcal{X}})}{\partial \mathbf{W}} \right\rangle = \mathbf{A}^\ell \mathbf{\Theta}^{(L)}(\bar{\mathcal{X}}, \bar{\mathcal{X}})\mathbf{A}^\ell. \quad (66)$$

Since the NTK matrix $\boldsymbol{\Theta}^{(L)}(\bar{\mathcal{X}}, \bar{\mathcal{X}})$ is permutation invariant with diagonal values being larger than non-diagonal values, we can write it as

$$\boldsymbol{\Theta}^{(L)}(\bar{\mathcal{X}}, \bar{\mathcal{X}}) = c'(\mathbf{I} + c\mathbf{1}\mathbf{1}^\top) \tag{67}$$

for some constants $c'$ and $c$ determined by the depth of the network $L$. Substituting it to (12) gives us the learning dynamics of infinitely-wide and arbitrarily-deep GNN

$$\left[\mathcal{R}_{t+1}, \mathcal{R}'_{t+1}\right] = -\eta\mathbf{A}^\ell(\mathbf{I} + c\mathbf{1}\mathbf{1}^\top)\mathbf{A}^\ell[\mathcal{R}_t, \mathbf{0}] + [\mathcal{R}_t, \mathcal{R}'_t]. \tag{68}$$

## B.6 PROOF OF COROLLARY 7: LINEAR GNN

For linear GNN defined as $[\mathcal{F}, \mathcal{F}'] = \mathbf{A}^\ell\bar{\mathcal{X}}\mathbf{W}$. The corresponding node-level GNTK $\boldsymbol{\Theta}_t^{(1)}$ for linear GNN is naturally constant and can be computed as

$$\begin{aligned}
\boldsymbol{\Theta}_t^{(1)}\left(\bar{\mathcal{X}}, \bar{\mathcal{X}}; \mathbf{A}\right) &= \nabla_\mathbf{W}[\mathcal{F}, \mathcal{F}']^\top \nabla_\mathbf{W}[\mathcal{F}, \mathcal{F}'] \\
&= \mathbf{A}^\ell\bar{\mathcal{X}}(\mathbf{A}^\ell\bar{\mathcal{X}})^\top.
\end{aligned} \tag{69}$$

**Case 1.** When the input $\bar{\mathcal{X}}$ is defined as an identity matrix, we have

$$\mathbf{A}^\ell\bar{\mathcal{X}}(\mathbf{A}^\ell\bar{\mathcal{X}})^\top = \mathbf{A}^{2\ell} \tag{70}$$

which is also a special case of Theorem 6 where the backbone MLP model is one layer and consequently $c = 0$. Based on (12), the learning dynamics of linear GNN can be written as

$$\left[\mathcal{R}_{t+1}, \mathcal{R}'_{t+1}\right] = -\eta\mathbf{A}^{2\ell}[\mathcal{R}_t, \mathbf{0}] + [\mathcal{R}_t, \mathcal{R}'_t] \tag{71}$$

which is equivalent to the basic version of RP in (6) with $K = 2\ell$.

**Case 2.** When the input $\bar{\mathcal{X}}$ is obtained from (full-rank) graph spectral decomposition, we have

$$\mathbf{A}^\ell\bar{\mathcal{X}}(\mathbf{A}^\ell\bar{\mathcal{X}})^\top = \mathbf{A}^\ell\bar{\mathcal{X}}\bar{\mathcal{X}}^\top\mathbf{A}^\ell = \mathbf{A}^{2\ell+1}. \tag{72}$$

In this case, the learning dynamics of linear GNN can be written as

$$\left[\mathcal{R}_{t+1}, \mathcal{R}'_{t+1}\right] = -\eta\mathbf{A}^{2\ell+1}[\mathcal{R}_t, \mathbf{0}] + [\mathcal{R}_t, \mathcal{R}'_t] \tag{73}$$

which is equivalent to the basic version of RP in (6) with $K = 2\ell + 1$. For non-PSD adjacency matrix $\mathbf{A}$, we also have the approximation $\mathbf{A}^\ell\bar{\mathcal{X}}^\top\bar{\mathcal{X}}\mathbf{A}^\ell \approx \mathbf{A}^{2\ell+1}$. However, in this case, the basic RP algorithm in (6) can not be implemented by the conventional (deep) learning framework (cf. Appendix 2).

## B.7 DERIVATION OF EQUATION 16: GENERALIZATION BOUND

Based on the Rademacher complexity-based generalization bound for kernel regression in (Bartlett & Mendelson, 2002), for training data $\{(\boldsymbol{x}_i, y_i)\}_{i=1}^{n_l}$ drawn i.i.d. from an underlying distribution $\mathcal{P}$, arbitrary loss function $l : \mathbb{R} \times \mathbb{R} \to [0, 1]$ that is 1-Lipschitz in the first argument such that $l(y, y) = 0$, we have that with probability at least $1 - \delta$, the population risk of kernel regression with respect to the limit NTK $\boldsymbol{\Theta}$ has upper bound (see proof in Bartlett & Mendelson (2002); Arora et al. (2019); Du et al. (2019b))

$$\mathbb{E}_{(\boldsymbol{x},y)\sim\mathcal{P}}\left[l\left(f(\boldsymbol{x}), y\right)\right] = O\left(\frac{\sqrt{\mathcal{Y}^\top\boldsymbol{\Theta}^{-1}\mathcal{Y} \cdot \text{Tr}(\boldsymbol{\Theta})}}{n_l} + \sqrt{\frac{\log(1/\delta)}{n_l}}\right). \tag{74}$$

We now assume an extreme case where the limit NTK matrix is determined by the graph, i.e. $\boldsymbol{\Theta} = (\mathbf{I} - \alpha\mathbf{A})^{-1} = \lim_{k\to\infty}\sum_{i=0}^k(\alpha\mathbf{A})^i$, which is equivalent to the propagation matrix adopted by the converged LP algorithm,[2] and is a valid kernel matrix by noting that the spectral radius $\rho(\mathbf{A}) \leq 1$ for normalized adjacency matrix $\mathbf{A}$ and $\alpha < 1$. For $\mathcal{Y}^\top\boldsymbol{\Theta}^{-1}\mathcal{Y}$, we have

$$\begin{aligned}
\mathcal{Y}^\top\boldsymbol{\Theta}^{-1}\mathcal{Y} &= \mathcal{Y}^\top\mathcal{Y} - \alpha\mathcal{Y}^\top\mathbf{A}\mathcal{Y} \\
&= n_l - \alpha\langle\mathcal{Y}\mathcal{Y}^\top, \mathbf{A}\rangle_F \\
&= n_l - cA(\boldsymbol{\Theta}^*, \mathbf{A})
\end{aligned} \tag{75}$$

---

[2]With slight abuse of notation, we denote by $\mathbf{A}$ the adjacency matrix for the training set, and $\boldsymbol{\Theta}^*$ the optimal kernel matrix for the training set.

where $c = \alpha \|\mathbf{\Theta}^*\|_F \|\mathbf{A}\|_F$ is a constant. For $\mathrm{Tr}(\mathbf{\Theta})$, we have

$$\mathrm{Tr}(\mathbf{\Theta}) = \sum_{i=1}^{n_l} \frac{1}{1 - \alpha \sigma_i(\mathbf{A})} \leq \frac{n_l}{1 - \alpha \sigma_{max}(\mathbf{A})} = O(n_l), \tag{76}$$

where $\sigma_{max}(\mathbf{A}) \leq 1$ is the maximal eigenvalue of $\mathbf{A}$. It follow that the population risk has upper bound

$$\mathbb{E}_{(\boldsymbol{x}, y) \sim \mathcal{P}} \left[ l\left(f(\boldsymbol{x}), y\right)\right] = O\left( \sqrt{\frac{n_l - cA(\mathbf{\Theta}^*, \mathbf{A})}{n_l}} + \sqrt{\frac{\log(1/\delta)}{n_l}} \right). \tag{77}$$

## B.8 PROOF OF THEOREM 9: BAYESIAN OPTIMALITY OF GNN

**Problem Setting.** In this proof, we consider the following setting. Suppose all possible samples are given by inputs $\mathcal{X} = \{\boldsymbol{x}_i\}_{i=1}^N$ and labels $\mathcal{Y} = \{y(\boldsymbol{x}_i)\}_{i=1}^N$ that are randomly generated from an unknown distribution $\mathcal{P}$, where $N$ could be arbitrarily large and both $\mathcal{X}$ and $\mathcal{Y}$ are assumed to be unobserved. For binary classification, we also have $y \in \{-1, 1\}$ with balanced label distribution, i.e. $\mathbb{E}[y] = 0$. Given that for an arbitrary pair of instances $\boldsymbol{x}$ and $\boldsymbol{x}'$, the probability that they share the same label is defined by a large matrix $\mathbf{A} \in \mathbb{R}^{N \times N}$:

$$P\left(y(\boldsymbol{x}) = y(\boldsymbol{x}')\right) = \mathbf{A}_{\boldsymbol{x}\boldsymbol{x}'}, \tag{78}$$

where $\mathbf{A}_{\boldsymbol{x}\boldsymbol{x}'}$ is an element in $\mathbf{A}$ that corresponds to sample pair $\boldsymbol{x}$ and $\boldsymbol{x}'$. Our task is to find the optimal kernel function whose corresponding kernel regression predictive function $f_{ker}(\cdot)$ minimizes the population risk

$$\mathbb{E}_{(\boldsymbol{x}, y) \sim \mathcal{P}} \left[ \left(f_{ker}(\boldsymbol{x}) - y(\boldsymbol{x})\right)^2 \right]. \tag{79}$$

**Proof.** Next, we give solution to the optimal kernel function that minimizes the population risk in the above setting. For data randomly generated from $\mathcal{P}$ and conditioned on (78), the random labels have covariance matrix $\mathbf{\Sigma}(\mathcal{X}, \mathcal{X}) \in \mathbb{R}^{N \times N}$ which satisfies

$$\begin{aligned}
\mathbf{\Sigma}(\boldsymbol{x}, \boldsymbol{x}') &= \mathbb{E}_{(\boldsymbol{x}, y) \sim \mathcal{P}} \left[ (y(\boldsymbol{x}) - \mathbb{E}[y])(y(\boldsymbol{x}') - \mathbb{E}[y]) \right] \\
&= \mathbb{E}_{(\boldsymbol{x}, y) \sim \mathcal{P}} \left[ y(\boldsymbol{x}) y(\boldsymbol{x}') \right] \\
&= 2\mathbf{A}_{\boldsymbol{x}\boldsymbol{x}'} - 1.
\end{aligned} \tag{80}$$

Let us consider kernel regression w.r.t. a kernel $\kappa$, whose corresponding predictive function $f_{ker}(\boldsymbol{x})$ on all possible unobserved samples $\mathcal{X}$ and $\mathcal{Y}$ is given by

$$f_{ker}(\boldsymbol{x}) = \kappa(\boldsymbol{x}, \mathcal{X}) \mathbf{K}(\mathcal{X}, \mathcal{X})^{-1} \mathcal{Y} = \mathbf{M}_{\boldsymbol{x}}^\top \mathcal{Y}. \tag{81}$$

where $\mathbf{M}_{\boldsymbol{x}} \in \mathbb{R}^N$ is a vector associated with sample $\boldsymbol{x}$ defined as $\mathbf{M}_{\boldsymbol{x}}^\top = \kappa(\boldsymbol{x}, \mathcal{X}) \mathbf{K}(\mathcal{X}, \mathcal{X})^{-1}$. Note that we do not differentiate training or testing samples here since all inputs and labels are assumed to be unseen. We aim to search the optimal kernel function that minimizes the population risk, which can be achieved if the risk for each sample $\boldsymbol{x}$ is minimized. Specifically, for an arbitrary sample $\boldsymbol{x}$, its risk is

$$\begin{aligned}
\mathbb{E}_y \left[ \left(f_{ker}(\boldsymbol{x}) - y(\boldsymbol{x})\right)^2 \right] &= \mathbb{E}_y \left[ \left(\mathbf{M}_{\boldsymbol{x}}^\top \mathcal{Y} - y(\boldsymbol{x})\right)^2 \right] \\
&= \mathbf{M}_{\boldsymbol{x}}^\top \mathbf{\Sigma}(\mathcal{X}, \mathcal{X}) \mathbf{M}_{\boldsymbol{x}} - 2\mathbf{M}_{\boldsymbol{x}}^\top \mathbf{\Sigma}(\mathcal{X}, \boldsymbol{x}) + \mathbf{\Sigma}(\boldsymbol{x}, \boldsymbol{x}).
\end{aligned} \tag{82}$$

To find the optimal kernel that minimizes the risk, we differentiate it w.r.t. $\mathbf{M}_{\boldsymbol{x}}$,

$$\nabla_{\mathbf{M}_{\boldsymbol{x}}} \mathbb{E}_{(\boldsymbol{x}, y) \sim \mathcal{P}} \left[ \left(f_{ker}(\boldsymbol{x}) - y(\boldsymbol{x})\right)^2 \right] = 2\mathbf{\Sigma}(\mathcal{X}, \mathcal{X}) \mathbf{M}_{\boldsymbol{x}} - 2\mathbf{\Sigma}(\mathcal{X}, \boldsymbol{x}) \tag{83}$$

which gives us the minimizer

$$\mathbf{M}_{\boldsymbol{x}}^\top = \mathbf{\Sigma}(\boldsymbol{x}, \mathcal{X}) \mathbf{\Sigma}(\mathcal{X}, \mathcal{X})^{-1}. \tag{84}$$

And since $\mathbf{M}_{\boldsymbol{x}}^\top = \kappa(\boldsymbol{x}, \mathcal{X}) \mathbf{K}(\mathcal{X}, \mathcal{X})^{-1}$, the population risk is minimized on each single data point for optimal kernel $\kappa^*$ whose kernel matrix is

$$\mathbf{K}^*(\mathcal{X}, \mathcal{X}) = 2\mathbf{A} - 1. \tag{85}$$

*Remark.* The above analysis also generalize to the setting when $\mathbf{A}$ is defined over a subset of all possible samples, in which case the optimal kernel matrix is still $2\mathbf{A} - 1$ but it is no longer possible to minimize population risk outside the coverage of $\mathbf{A}$.

## C ADDITIONAL DISCUSSIONS

### C.1 DIFFERENT SETTINGS: TRANSDUCTIVE, INDUCTIVE AND LEARNING WITHOUT GRAPH

Transductive (semi-supervised) and inductive (supervised) learning are two types of common settings in node-level classification tasks. The former incorporates unlabeled nodes (testing samples) in the training process while the latter only has access to unlabeled nodes for inference. Recall that the residual propagation process of general parameterized models with arbitrary unseen testing samples can be written as

$$
[\mathcal{R}_{t+1}, \mathcal{R}'_{t+1}] = -\eta \begin{bmatrix} \underbrace{\Theta_t(\mathcal{X}, \mathcal{X})}_{n_l \times n_l} & \Theta_t(\mathcal{X}, \mathcal{X}') \\ \underbrace{\Theta_t(\mathcal{X}', \mathcal{X})}_{(n-n_l) \times n_l} & \Theta_t(\mathcal{X}', \mathcal{X}') \end{bmatrix} \cdot \underbrace{[\mathcal{R}_t, \mathbf{0}]}_{n} + \underbrace{[\mathcal{R}_t, \mathcal{R}'_t]}_{n}. \tag{86}
$$

For node-level tasks, the difference between transductive and inductive settings boils down to different feature map of the kernel function for training and testing sets. To be specific:

**Transductive learning.** For transductive learning or semi-supervised learning, the residual propagation can be written as

$$
[\mathcal{R}_{t+1}, \mathcal{R}'_{t+1}] = -\eta \begin{bmatrix} \underbrace{\Theta_t(\mathcal{X}, \mathcal{X}; \mathbf{A})}_{n_l \times n_l} & \Theta_t(\mathcal{X}, \mathcal{X}'; \mathbf{A}) \\ \underbrace{\Theta_t(\mathcal{X}', \mathcal{X}; \mathbf{A})}_{(n-n_l) \times n_l} & \Theta_t(\mathcal{X}', \mathcal{X}'; \mathbf{A}) \end{bmatrix} \cdot \underbrace{[\mathcal{R}_t, \mathbf{0}]}_{n} + \underbrace{[\mathcal{R}_t, \mathcal{R}'_t]}_{n}. \tag{87}
$$

where $\Theta_t(\mathcal{X}, \mathcal{X}; \mathbf{A})$ is the node-level GNTK defined in Sec. 4. This equation is equivalent to the one presented in the main text, i.e. (12).

**Inductive learning.** Let us denote $\mathbf{A}_{\mathcal{X}\mathcal{X}}$ the submatrix of $\mathbf{A}$ corresponding to the training set. For inductive learning or supervised learning, the residual propagation can be written as

$$
[\mathcal{R}_{t+1}, \mathcal{R}'_{t+1}] = -\eta \begin{bmatrix} \underbrace{\Theta_t(\mathcal{X}, \mathcal{X}; \mathbf{A}_{\mathcal{X}\mathcal{X}})}_{n_l \times n_l} & \Theta_t^{(ind)}(\mathcal{X}, \mathcal{X}'; \mathbf{A}) \\ \underbrace{\Theta_t^{(ind)}(\mathcal{X}', \mathcal{X}; \mathbf{A})}_{(n-n_l) \times n_l} & \Theta_t^{(ind)}(\mathcal{X}', \mathcal{X}'; \mathbf{A}) \end{bmatrix} \cdot \underbrace{[\mathcal{R}_t, \mathbf{0}]}_{n} + \underbrace{[\mathcal{R}_t, \mathcal{R}'_t]}_{n}, \tag{88}
$$

Let us denote predictions (for the training set) given by the GNN model with matrix $\mathbf{A}_{\mathcal{X}\mathcal{X}}$ as $\mathcal{F}_{ind}$, and the predictions (for the testing set) given by the GNN model with matrix $\mathbf{A}$ as $\mathcal{F}'_{trans}$, we have

$$
\begin{aligned}
\Theta_t(\mathcal{X}, \mathcal{X}; \mathbf{A}_{\mathcal{X}\mathcal{X}}) &= \nabla_{\mathbf{W}}\mathcal{F}_{ind}^{\top}\nabla_{\mathbf{W}}\mathcal{F}_{ind}, \\
\Theta_t^{(ind)}(\mathcal{X}', \mathcal{X}; \mathbf{A}) &= \nabla_{\mathbf{W}}\mathcal{F}'^{\top}_{trans}\nabla_{\mathbf{W}}\mathcal{F}_{ind}.
\end{aligned} \tag{89}
$$

In this case, the $n \times n$ matrix is still a valid kernel matrix with graph inductive bias (cf. Appendix B.3), and thus our insights from the transductive setting still holds.

**Training without graph.** Another interesting setting studied recently (Yang et al., 2023) is using vanilla MLP for training and then adopts the GNN architecture in inference, which leads to the so-called PMLP that can accelerate training without deteriorating the generalization performance. In this setting, the residual propagation can be written as

$$
[\mathcal{R}_{t+1}, \mathcal{R}'_{t+1}] = -\eta \begin{bmatrix} \underbrace{\Theta_t(\mathcal{X}, \mathcal{X})}_{n_l \times n_l} & \Theta_t^{(pmlp)}(\mathcal{X}, \mathcal{X}'; \mathbf{A}) \\ \underbrace{\Theta_t^{(pmlp)}(\mathcal{X}', \mathcal{X}; \mathbf{A})}_{(n-n_l) \times n_l} & \Theta_t^{(pmlp)}(\mathcal{X}', \mathcal{X}'; \mathbf{A}) \end{bmatrix} \cdot \underbrace{[\mathcal{R}_t, \mathbf{0}]}_{n} + \underbrace{[\mathcal{R}_t, \mathcal{R}'_t]}_{n}, \tag{90}
$$

where $\mathbf{\Theta}_t(\mathcal{X}, \mathcal{X})$ is equivalent to the NTK matrix of fully-connected neural networks. This is equivalent to the inductive setting where the graph adjacency matrix used in training is an identity matrix: let us denote predictions (for the training set) given by MLP as $\mathcal{F}_{mlp}$, we have

$$\mathbf{\Theta}_t(\mathcal{X}, \mathcal{X}) = \nabla_{\mathbf{W}} \mathcal{F}_{mlp}^\top \nabla_{\mathbf{W}} \mathcal{F}_{mlp},$$

$$\mathbf{\Theta}_t^{(ind)}(\mathcal{X}', \mathcal{X}; \mathbf{A}) = \nabla_{\mathbf{W}} {\mathcal{F}'}_{trans}^\top \nabla_{\mathbf{W}} \mathcal{F}_{mlp}. \tag{91}$$

The success of PMLP is consistent with the insight that the residual flow between training and testing sets with graph inductive bias is important to explain the generalization of GNNs.

## C.2 OTHER LOSS FUNCTIONS

It is worth noting that residual propagation process is not exclusive to the squared loss. While our analysis focuses on the squared loss, similar residual propagation schemes can be obtained with other loss functions that will change the original linear residual propagation process to non-linear. Therefore, the insight still holds for other loss functions.

**Mean squared error.** For completeness, let us rewrite the residual propagation process for general parameterized model (e.g. linear model, fully-connected neural network, GNN) with squared loss here, with derivation given in Appendix B.1

$$\left[\mathcal{R}_{t+1}, \mathcal{R}'_{t+1}\right] = -\eta\, \mathbf{\Theta}_t(\bar{\mathcal{X}}, \bar{\mathcal{X}})[\mathcal{R}_t, \mathbf{0}] \;+\; [\mathcal{R}_t, \mathcal{R}'_t], \tag{92}$$

which is a linear propagation process, where $\mathbf{\Theta}_t(\bar{\mathcal{X}}, \bar{\mathcal{X}}) = \nabla_{\mathbf{W}}[\mathcal{F}_t, \mathcal{F}'_t]^\top \nabla_{\mathbf{W}}[\mathcal{F}_t, \mathcal{F}'_t] \in \mathbb{R}^{n \times n}$.

**Mean absolute error.** If the loss function is a MAE (a.k.a. $L_1$) loss $\mathcal{L} = \|\mathcal{F}_t - \mathcal{Y}\|_1$, then the residual propagation process can be revised as

$$\left[\mathcal{R}_{t+1}, \mathcal{R}'_{t+1}\right] = -\eta\, \mathbf{\Theta}_t(\bar{\mathcal{X}}, \bar{\mathcal{X}})[\mathrm{sgn}(\mathcal{R}_t), \mathbf{0}] \;+\; [\mathcal{R}_t, \mathcal{R}'_t], \tag{93}$$

where $\mathrm{sgn}(x)$ is the sign function that output 1 is $x > 0$ otherwise $-1$, and could be thought of as coarsening the residual information to a binary value.

**Cross-entropy.** For the CE loss commonly used for classification tasks, we define the residual as $\mathcal{R}_t = \mathcal{Y} - \sigma \mathcal{F}_t$, where $\sigma$ here denotes sigmoid activation and applies on every element in vector $\mathcal{F}_t$. The residual propagation process can be revised as

$$\left[\mathcal{R}_{t+1}, \mathcal{R}'_{t+1}\right] = -\eta\, \mathbf{\Theta}_t(\bar{\mathcal{X}}, \bar{\mathcal{X}})[c(\mathcal{R}_t), \mathbf{0}] \;+\; [\mathcal{R}_t, \mathcal{R}'_t], \tag{94}$$

where the function $c(\cdot)$ denotes multiplying a scaling factor $1/(1 - \sigma f_t(\boldsymbol{x}_i))\sigma f_t(\boldsymbol{x}_i)$ to each residual for re-weighting. This scaling factor up-weights smaller residuals whose corresponding predictions are more confident, and down-weights larger residuals whose predictions are less confident.

## C.3 MULTI-DIMENSIONAL OUTPUT

In this section, we discuss the modification of analysis for learning dynamics of GNNs when outputs $f(\boldsymbol{x}) \in \mathbb{R}^c$ is multi-dimensional. For the multi-dimensional output case where $f(\boldsymbol{x}) \in \mathbb{R}^c$ and $c$ is the dimension of outputs, let $\mathcal{F}_t \in \mathbb{R}^{n_l c \times 1}$ and $\mathcal{R}_t \in \mathbb{R}^{n_l c \times 1}$ be model prediction and residuals in vector forms. Then, the residual dynamics for the training set (which can straight-forwardly incorporate

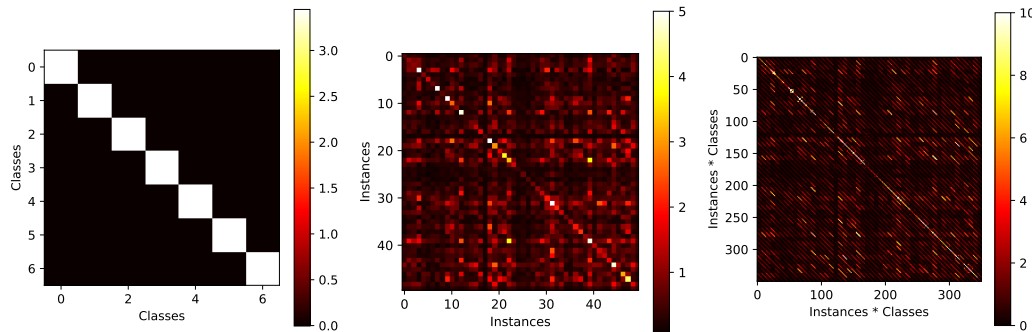

Figure 4: Visualization of NTK of well-trained GCN on a node classification benchmark (Cora). 50 nodes are randomly selected for clearity. From left to right are $c \times c$, $n \times n$, $nc \times nc$ NTK matrices, where the former two matrices are obtained by averaging the $nc \times nc$ NTK matrix at dimension $n$ and $c$ respectively. The diagonal patterns in the first and last matrix verifies that our analysis for finitely-wide GNNs in binary classification also applies to multi-class classification setting.

testing samples similar to the derivation in Appendix B.1) is revised as

$$\frac{\partial \mathcal{R}_t}{\partial t} = \frac{\partial \mathcal{R}_t}{\partial \mathbf{W}_t} \cdot \frac{\partial \mathbf{W}_t}{\partial t} \qquad \text{(Chain rule)} \qquad (95)$$

$$= -\eta \cdot \frac{\partial \mathcal{R}_t}{\partial \mathbf{W}_t} \cdot \nabla_{\mathbf{W}} \mathcal{L} \qquad \text{(GD training)} \qquad (96)$$

$$= -\eta \cdot \frac{\partial \mathcal{R}_t}{\partial \mathbf{W}_t} \cdot \nabla_{\mathbf{W}} \mathcal{F}_t \cdot \nabla_f \mathcal{L} \qquad \text{(Chain rule)} \qquad (97)$$

$$= \eta \cdot \underbrace{\nabla_{\mathcal{W}}^{\top} \mathcal{F}_t}_{n_l c \times |\mathbf{W}|} \cdot \underbrace{\nabla_{\mathbf{W}} \mathcal{F}_t}_{|\mathbf{W}| \times n_l c} \cdot \underbrace{\nabla_f \mathcal{L}}_{n_l c \times 1} \qquad \text{(Change of notation)} \qquad (98)$$

$$= \eta \cdot \underbrace{\mathbf{\Theta}_t(\mathcal{X}, \mathcal{X})}_{n_l c \times n_l c} \cdot \underbrace{\nabla_f \mathcal{L}}_{n_l c \times 1} \qquad (99)$$

$$= -\eta \cdot \underbrace{\mathbf{\Theta}_t(\mathcal{X}, \mathcal{X})}_{n_l c \times n_l c} \cdot \underbrace{\mathcal{R}_t}_{n_l c \times 1} \qquad \text{(Squared loss)}. \qquad (100)$$

Compared with residual propagation process for scalar output, the residual propagation process for multi-dimensional output additionally incorporate flow of residual across different dimensions. However, for infinite neural networks, the NTK $\mathbf{\Theta}(\mathcal{X}, \mathcal{X})$ for multi-dimensional output (Jacot et al., 2018) can be written as

$$\underbrace{\mathbf{\Theta}(\mathcal{X}, \mathcal{X})}_{n_l c \times n_l c} = \underbrace{\bar{\mathbf{\Theta}}(\mathcal{X}, \mathcal{X})}_{n_l \times n_l} \otimes \underbrace{\mathbf{I}_c}_{c \times c}. \qquad (101)$$

Namely, there is no cross-dimension flow in residual propagation and our analysis for scalar output can be trivially adapted to the multi-dimensional output case. For finite neural networks, while cross-dimension residual propagation is theoretically inevitable, one can still resort the the above decomposition as approximation. For example, we can define pseudo NTK as

$$\bar{\mathbf{\Theta}}_t(\mathcal{X}, \mathcal{X}) = \left( \nabla_{\mathbf{W}} \sum_{h=1}^{c} [\mathcal{F}_t]_h \right)^{\top} \left( \nabla_{\mathbf{W}} \sum_{h=1}^{c} [\mathcal{F}_t]_h \right), \qquad (102)$$

and recent works (Mohamadi & Sutherland, 2023) has proved that such a kernel function can be used to approximate the original $n_l c \times n_l c$ kernel matrix:

$$\frac{\left\| \bar{\mathbf{\Theta}}_t(\boldsymbol{x}_i, \boldsymbol{x}_j) \otimes \mathbf{I}_c - \mathbf{\Theta}_t(\boldsymbol{x}_i, \boldsymbol{x}_j) \right\|_F}{\left\| \mathbf{\Theta}_t(\boldsymbol{x}_i, \boldsymbol{x}_j) \right\|_F} \in \tilde{\mathcal{O}}\left( n_l^{-\frac{1}{2}} \right). \qquad (103)$$

To empirically verify this for finitely-wide GNNs, we visualize the NTK matrix for real-world GNNs in Cora dataset in Fig. 4. Therefore, our insights for both infinitely-wide GNNs and finitely-wide GNNs still hold in multi-dimensional case, which are also verified by our experiments.

Table 3: Statistics of 17 datasets.

| Type | Dataset | # Nodes ($n$) | # Edges ($e$) | # Features ($d$) | # Class ($c$) | # Tasks | Split |
|------|---------|---------------|---------------|------------------|---------------|---------|-------|
| OGB | Arxiv | 169,343 | 1,166,243 | 128 | 40 | 1 | Public |
| | Proteins | 132,534 | 39,561,252 | 8 | 2 | 112 | Public |
| | Products | 2,449,029 | 61,859,140 | 100 | 47 | 1 | Public |
| Homophilic | Cora | 2,708 | 10,556 | 1,433 | 7 | 1 | Public |
| | Citeseer | 3,327 | 9,104 | 3,703 | 6 | 1 | Public |
| | Pubmed | 19,717 | 88,648 | 500 | 3 | 1 | Public |
| | Computers | 13,752 | 491,722 | 767 | 10 | 1 | 80%/10%/10% |
| | Photo | 7,650 | 238,162 | 745 | 8 | 1 | 80%/10%/10% |
| | CS | 18,333 | 163,788 | 6,805 | 15 | 1 | 80%/10%/10% |
| | Physics | 34,493 | 495,924 | 8,415 | 5 | 1 | 80%/10%/10% |
| Heterophilic | roman-empire | 22662 | 32927 | 300 | 18 | 1 | Public |
| | amazon-ratings | 24,492 | 93,050 | 300 | 5 | 1 | Public |
| | minesweeper | 10,000 | 39,402 | 7 | 2 | 1 | Public |
| | tolokers | 11,758 | 519,000 | 10 | 2 | 1 | Public |
| | questions | 48,921 | 153,540 | 301 | 2 | 1 | Public |
| | Texas | 183 | 325 | 1,703 | 5 | 1 | Public |
| Synthetic | Synthetic | 2,000 | $8,023 \sim 8,028$ | 100 | 5 | 1 | 100/500/1000 |

# D  IMPLEMENTATION DETAILS

## D.1  EXPERIMENTS IN SECTION 3.2

`Arxiv`, `Proteins` and `Products` (Hu et al., 2020) are three relatively large datasets containing 169343, 132534 and 2449029 nodes respectively.

- The `Arxiv` dataset represents the citation network between all computer science arxiv papers. Each node is associated with a feature vector representing the averaged embeddings of words in the title and abstract of that paper and the task is to predict the subject areas.
- For `Proteins` dataset, nodes represent proteins and edges represent biologically significant associations between proteins, categorized by their types. The task is to predict the presence or absence of 112 protein functions as a multi-label binary classification problem.
- The `Products` dataset is an Amazon product co-purchasing network where nodes are products in Amazon and links represent two products are purchased together. The feature for each node is dimensionality-reduced bag-of-words for the product descriptions and the task is predict the category of a product.

We follow the original splitting of Hu et al. (2020) for evaluation. The statistics of these datasets are shown in Table 3.

We compare GRP with several classic methods for learning on graphs: standard *MLP*, Label Propagation (*LP*) (Zhu et al., 2003), *LinearGNN* (SGC) (Wu et al., 2019), *GNN (GCN)* (Kipf & Welling, 2017). Except the results of linear GNN which are from our reproduction, the results of other baselines align with the results reported in the OGB leaderboard, where the detailed implementation and hyperparameter setup can be found. [3] The LP algorithm reported in Tab. 1 follows the standard implementation that is ran until convergence, while the LP algorithm reported in Fig. 2 does not run until convergence in order to align with the proposed RP algorithm.

For hyperparamter search of RP, we adopt grid search for the RP algorithm with the step size $\eta$ from $\{0.01, 0.02, 0.05, 0.1, 0.2, 0.5, 1\}$, the power $K$ ranging from 1 to 10. For `Arxiv`, `Proteins` and `Products`, $K$ is chosen as 7, 1, 8 respectively. Since both LP and GRP are deterministic algorithms, their standard deviations are 0. All experiments are conducted on Quadro RTX 8000 with 48GB memory.

## D.2  EXPERIMENTS IN SECTION 5.2

We conduct experiments on real-world benchmark datasets `Cora` and `Texas`, and a synthetic dataset generated by the stochastic block model. For the synthetic dataset, we set number of blocks as 5 (i.e.

---

[3] https://ogb.stanford.edu/docs/leader_nodeprop

number of classes) with each block having $400$ nodes. Each node is associated with a $100$-dimensional informative input feature vector. For the homophilic version of the dataset, the $5 \times 5$ edge probability matrix is defined as $\mathbf{P} = 0.01 \cdot \mathbf{I}_5$, i.e. nodes in the same block are connected with probability $0.01$, and we gradually change this matrix in the generation process until there are only heterophilic edges left in the dataset, i.e. $\mathbf{P} = 0.0025 \cdot \mathbf{1}\mathbf{1}^\top - 0.0025 \cdot \mathbf{I}_5$. The statistics of these datasets are shown in Table 3. Note that we do not consider the large-scale datasets as in Section 3.2, since the computing NTK matrix is extremely costly in memory, especially for GNNs in node-level tasks where the output for an instance is also related to input features of other instances and mini-batch partitioning can not be directly adopted.

For the model, we choose a two-layer GCN with width $16$, bias term and ReLU activation for all datasets. In order to control the variable, when comparing NTK of the model using different graph structures, we fix the weights in the model, which is achieved by training on a fixed graph structure but evaluate the NTK matrix (and test performance) using different graph structure. For real-world datasets, we use the original graph for training which is equivalent to the standard training, and for synthetic dataset, we use an identity matrix for training in which case the model is equivalent to the recently proposed PMLP (Yang et al., 2023) model that has shown to as effective as GNNs in the supervised learning setting. The optimization algorithm is gradient descent with learning rates $1e-2$, $3e-4$, $5e-5$ respectively for `Cora`, `Texas`, `Synthetic` respectively, momentum $0.9$ and weight decay $5e-4$. The loss function is the standard cross-entropy loss for multi-class classification.

Since the NTK matrix for multi-dimensional output is a $nc \times nc$ matrix, where $c$ is the output dimension, we follow prior work (see Mohamadi & Sutherland (2023) and references therein) and compute the $n \times n$ NTK matrix by averaging over the dimension $c$. The graph adjacency matrix we consider here is defined as $\mathbf{A}^4$ in order to align with our theoretical result for two-layer infinitely-wide GNN in Theorem 5. We also normalize these matrices before computing the alignment following (Cortes et al., 2012; Baratin et al., 2021) as a standard way of preprocess.

### D.3 Algorithm Description

Recall the basic version of RP can be described by the following iterative forward propagation process:

$$\left[\mathcal{R}_{t+1}, \mathcal{R}'_{t+1}\right] = -\eta \mathbf{A}^K [\mathcal{R}_t, \mathbf{0}] + [\mathcal{R}_t, \mathcal{R}'_t]. \tag{104}$$

The intial conditions are

$$[\mathcal{R}_0, \mathcal{R}'_0] = [\mathcal{Y} - \mathcal{F}_0, \mathbf{0} - \mathcal{F}'_0], \tag{105}$$

where $\mathcal{F}_0 = \mathbf{0}$ and $\mathcal{F}'_0 = \mathbf{0}$. For real-world applications where the outputs are usually multi-dimensional, say, multi-class classification with $c$ classes, we define: $\mathcal{Y} = \{\boldsymbol{y}_i\}_{i=1}^{n_l} \in \mathbb{R}^{n_l \times c}$ where $\boldsymbol{y}_i$ is a onehot vector, $\mathbf{A} \in \mathbb{R}^{n \times n}$ is the same standard normalized adjacency matrix in GCN (Kipf & Welling, 2017). The pseudo code for the basic version of RP are shown in the following algorithm.

---

**Algorithm 1:** Basic version of residual propagation.

---

**Input:** Raw graph adjacency matrix $\mathbf{A}$, ground-truth labels for training samples $\mathcal{Y}$, step size $\eta$, power $K$.

Compute the normalized graph adjacency matrix by $\mathbf{A} \leftarrow \mathbf{D}^{-\frac{1}{2}}(\mathbf{A} + \mathbf{I})\mathbf{D}^{-\frac{1}{2}}$

Initialize $\mathcal{R}_0 \leftarrow \mathcal{Y} \in \mathbb{R}^{n_l \times c}$

Initialize $\mathcal{R}'_0 \leftarrow \mathbf{0} \in \mathbb{R}^{(n-n_l) \times c}$

**while** *Validation performance increases* **do**

    Label/residual propagation on the graph:

    $\tilde{\mathcal{R}}_{t-1} \leftarrow [\mathcal{R}_{t-1}, \mathbf{0}] \in \mathbb{R}^{n \times c}$

    **for** $i \leftarrow 1$ *to* $K$ **do**

        $\tilde{\mathcal{R}}_{t-1} \leftarrow \mathbf{A}\,\tilde{\mathcal{R}}_{t-1}$

    Update residuals $[\mathcal{R}_t, \mathcal{R}'_t] \leftarrow [\mathcal{R}_{t-1}, \mathcal{R}'_{t-1}] - \eta\tilde{\mathcal{R}}_{t-1}$

Output prediction for testing samples $\mathcal{F}' = -\mathcal{R}'$

---

# E ADDITIONAL EXPERIMENTS

## E.1 INCORPORATING INPUT FEATURES

For smaller datasets where node features are often more useful (i.e. the dimension of node features is closer to the size of dataset, for example in `Citeseer`, the node feature dimension is even larger than the size of dataset), we consider the following generalized RP algorithm that combines kernel methods to leverage node feature information in the propagation process

$$\left[\mathcal{R}_{t+1}, \mathcal{R}'_{t+1}\right] = \left(\mathbf{I}_n - \eta \mathbf{A}^K \mathbf{K}(\bar{\mathcal{X}}, \bar{\mathcal{X}}) \mathbf{A}^K\right) \left[\mathcal{R}_t, \mathbf{0}\right] + \left[\mathbf{0}, \mathcal{R}'_t\right], \quad \text{where } \mathcal{R}_0 = \mathcal{Y}, \mathcal{R}'_0 = \mathbf{0}, \quad (106)$$

where $\mathbf{K}(\bar{\mathcal{X}}, \bar{\mathcal{X}}) \in \mathbb{R}^{n \times n}$ could be specified as arbitrary kernel functions (such as Sigmoid kernel, Gaussian kernel, etc.) that is applied to compute pairwise similarities. This variant of RP could also be treated as propagation on kernel's RKHS, i.e.

$$\mathbf{A}^K \mathbf{K}(\bar{\mathcal{X}}, \bar{\mathcal{X}}) \mathbf{A}^K = (\mathbf{A}^K \mathbf{K}(\bar{\mathcal{X}}, \cdot))(\mathbf{A}^K \mathbf{K}(\bar{\mathcal{X}}, \cdot))^\top, \quad (107)$$

which is impossible to directly implement in practice but can be achieved by our proposed RP.

With this implementation, RP can still run efficiently by treating the computation of the kernel matrix as a part of data prepossessing. Specifically, we will test Gaussian kernel which is defined as

$$\mathbf{K}(\boldsymbol{x}_i, \boldsymbol{x}_j) = \exp\left(-\frac{\|\boldsymbol{x}_i - \boldsymbol{x}_j\|^2}{2\sigma^2}\right), \quad (108)$$

while in practice one can treat the kernel function as a hyperparameter to tune for even better performance. The pseudo code for this version of RP are shown in the following algorithm.

---

**Algorithm 2:** Generalized residual propagation with kernel functions.

---

**Input:** Raw graph adjacency matrix $\mathbf{A}$, ground-truth labels for training samples $\mathcal{Y}$, input features $\mathcal{X}$ and $\mathcal{X}'$, step size $\eta$, power $K$.

Compute the normalized graph adjacency matrix by $\mathbf{A} \leftarrow \mathbf{D}^{-\frac{1}{2}}(\mathbf{A} + \mathbf{I})\mathbf{D}^{-\frac{1}{2}}$

Compute the kernel matrix based on input features $\mathbf{K}([\mathcal{X}, \mathcal{X}'], [\mathcal{X}, \mathcal{X}'])$

Initialize $\mathcal{R}_0 \leftarrow \mathcal{Y} \in \mathbb{R}^{n_l \times c}$

Initialize $\mathcal{R}'_0 \leftarrow \mathbf{0} \in \mathbb{R}^{(n-n_l) \times c}$

**while** *Validation performance increases* **do**

    Label/residual propagation on the graph:

    $\tilde{\mathcal{R}}_{t-1} \leftarrow [\mathcal{R}_{t-1}, \mathbf{0}] \in \mathbb{R}^{n \times c}$

    **for** $i \leftarrow 1$ *to* $K$ **do**

        $\tilde{\mathcal{R}}_{t-1} \leftarrow \mathbf{A}\,\tilde{\mathcal{R}}_{t-1}$

    $\tilde{\mathcal{R}}_{t-1} \leftarrow \mathbf{K}([\mathcal{X}, \mathcal{X}'], [\mathcal{X}, \mathcal{X}'])\,\tilde{\mathcal{R}}_{t-1}$

    **for** $i \leftarrow 1$ *to* $K$ **do**

        $\tilde{\mathcal{R}}_{t-1} \leftarrow \mathbf{A}\,\tilde{\mathcal{R}}_{t-1}$

    Update residuals $[\mathcal{R}_t, \mathcal{R}'_t] \leftarrow [\mathcal{R}_{t-1}, \mathcal{R}'_{t-1}] - \eta \tilde{\mathcal{R}}_{t-1}$

Output prediction for testing samples $\mathcal{F}' = -\mathcal{R}'$

---

## E.2 HOMOPHILIC DATASETS

To evaluate the generalized RP algorithm in (106), we experiment on 7 more (smaller) datasets: `Cora`, `Citeseer`, `Pubmed`, `Computer`, `Photo`, `CS`, `Physics`. For `Cora`, `Citeseer`, `Pubmed`, we follow the public split, while for other datasets, we randomly split them into training/validation/testing sets based on ratio 8/1/1. Statistics of these datasets are reported in Table. 3. For RP, we tune the hyperparameters $K$ and $\sigma$. For baselines (MLP, SGC Wu et al. (2019), GCN (Kipf & Welling, 2017), JKNet Xu et al. (2018), APPNP Klicpera et al. (2019)), we tune the hyperparameters provided in their original paper and report mean and standard deviation of testing accuracy with 20 different runs.

The results are reported in Table. 4. We found the proposed RP almost always achieves the best or second best performance, and outperforms GCN in 6 out of 7 datasets even using no learnable parameters. In terms of the average performance, RP achieved the highest ranking out of all popular GNN models considered. Better performance can potentially be achieved by considering more advanced kernel functions or other specialized propagation matrices.

Table 4: Performance of generalized RP on homophilic datasets. We mark the first and second place with gold and silver, and compare its performance with GCN by $\Delta_{GCN}$.

| Model | Cora | Citeseer | Pubmed | Computer | Photo | CS | Physics | Avg. |
|---|---|---|---|---|---|---|---|---|
| MLP | $59.7 \pm 1.0$ | $57.1 \pm 0.5$ | $68.4 \pm 0.5$ | $85.42 \pm 0.51$ | $92.91 \pm 0.48$ | $95.97 \pm 0.22$ | $96.90 \pm 0.27$ | 79.49 |
| SGC | $81.0 \pm 0.5$ | $71.9 \pm 0.5$ | $78.9 \pm 0.4$ | $89.92 \pm 0.37$ | $94.35 \pm 0.19$ | $94.00 \pm 0.30$ | $96.19 \pm 0.13$ | 86.61 |
| GCN | $81.9 \pm 0.5$ | $71.6 \pm 0.4$ | $79.3 \pm 0.3$ | $92.25 \pm 0.61$ | $95.16 \pm 0.92$ | $94.10 \pm 0.34$ | $96.64 \pm 0.36$ | 87.28 |
| JKNet | $81.3 \pm 0.5$ | $69.7 \pm 0.2$ | $78.9 \pm 0.6$ | $91.25 \pm 0.76$ | $94.82 \pm 0.22$ | $93.57 \pm 0.49$ | $96.31 \pm 0.29$ | 86.55 |
| APPNP | $82.6 \pm 0.2$ | $71.7 \pm 0.5$ | $80.3 \pm 0.1$ | $91.81 \pm 0.78$ | $95.84 \pm 0.34$ | $94.41 \pm 0.29$ | $96.84 \pm 0.26$ | 87.64 |
| RP (Ours) | $82.7 \pm 0.0$ | $73.0 \pm 0.0$ | $80.1 \pm 0.0$ | $92.00 \pm 0.00$ | $95.55 \pm 0.00$ | $94.60 \pm 0.00$ | $96.75 \pm 0.00$ | 87.81 |
| $\Delta_{GCN}$ | + 0.8 | + 1.4 | + 1.2 | - 0.25 | + 0.39 | + 0.19 | + 0.11 | + 0.53 |

Table 5: Performance of generalized RP on heterophilic datasets. The first three datasets use Accuracy, and the last two datasets use ROC-AUC. We report the ranking of RP among all baselines.

| Model | roman-empire | amazon-ratings | minesweeper | tolokers | questions | Avg. |
|---|---|---|---|---|---|---|
| ResNet | $65.88 \pm 0.38$ | $45.90 \pm 0.52$ | $50.89 \pm 1.39$ | $72.95 \pm 1.06$ | $70.34 \pm 0.76$ | 61.39 |
| H2GCN (Zhu et al., 2020) | $60.11 \pm 0.52$ | $36.47 \pm 0.23$ | $89.71 \pm 0.31$ | $73.35 \pm 1.01$ | $63.59 \pm 1.46$ | 64.64 |
| CPGNN (Zhu et al., 2021) | $63.96 \pm 0.62$ | $39.79 \pm 0.77$ | $52.03 \pm 5.46$ | $73.36 \pm 1.01$ | $65.96 \pm 1.95$ | 59.02 |
| GPR-GNN (Chien et al., 2021) | $64.85 \pm 0.27$ | $44.88 \pm 0.34$ | $86.24 \pm 0.61$ | $72.94 \pm 0.97$ | $55.48 \pm 0.91$ | 64.88 |
| FSGNN (Maurya et al., 2022) | $79.92 \pm 0.56$ | $52.74 \pm 0.83$ | $90.08 \pm 0.70$ | $82.76 \pm 0.61$ | $78.86 \pm 0.92$ | 76.87 |
| GloGNN (Li et al., 2022) | $59.63 \pm 0.69$ | $36.89 \pm 0.14$ | $51.08 \pm 1.23$ | $73.39 \pm 1.17$ | $65.74 \pm 1.19$ | 57.35 |
| FAGCN (Bo et al., 2021) | $65.22 \pm 0.56$ | $44.12 \pm 0.30$ | $88.17 \pm 0.73$ | $77.75 \pm 1.05$ | $77.24 \pm 1.26$ | 70.50 |
| GBK-GNN (Du et al., 2022) | $74.57 \pm 0.47$ | $45.98 \pm 0.71$ | $90.85 \pm 0.58$ | $81.01 \pm 0.67$ | $74.47 \pm 0.86$ | 73.58 |
| JacobiConv (Wang & Zhang, 2022) | $71.14 \pm 0.42$ | $43.55 \pm 0.48$ | $89.66 \pm 0.40$ | $68.66 \pm 0.65$ | $73.88 \pm 1.16$ | 69.38 |
| RP (Ours) | $66.01 \pm 0.56$ | $47.95 \pm 0.57$ | $80.48 \pm 0.76$ | $78.05 \pm 0.90$ | $76.39 \pm 1.16$ | 69.78 |
| **Rank** | **4 / 10** | **2 / 10** | **7 / 10** | **3 / 10** | **3 / 10** | **4 / 10** |

## E.3 HETEROPHILIC DATASETS

For a comprehensive evaluation, we further consider 5 heterophilic benchmarks `roman-empire`, `amazon-ratings`, `roman-empire`, `roman-empire`, `roman-empire` from a recent paper Platonov et al. (2023), which have addressed some drawbacks of standard datasets used for evaluating heterophily-specific models. We use 10 existing standard train/validation/test splits provided in their paper, and statistics of these datasets are also reported in Table. 3. Baselines are recently proposed strong GNN models that are carefully designed to tackle the heterophily problem.

The results are reported in Table. 5. We observed that the proposed RP maintains a strong level of performance when compared to these meticulously designed models, surpassing 6 out of the 9 in terms of average performance. It is important to note that RP has been proven to be suboptimal when applied to heterophilic graphs and is orthogonal to various other techniques designed for addressing this challenge. These observations suggest that there is significant untapped potential for further improvement of the algorithm to address this issue.

