# OpenReview forum: "How Graph Neural Networks Learn: Lessons from Training Dynamics in Function Space"
_ICLR.cc/2024/Conference — Submitted to ICLR 2024_

### Official Review · Reviewer_ZpWJ · 2023-10-31

**Soundness:** 4 excellent
**Presentation:** 4 excellent
**Contribution:** 4 excellent
**Rating:** 8
**Confidence:** 3

**Summary:**

This paper analyzes the learning dynamics of GNN in the function space and connects it to label propagation. The link is the residual propagation where the neural tangent kernel matrix is replaced by high order graph adjacency matrix. The authors show that the learning dynamics of infinitely wide two-layer GNN is a special form of residual propagation. The authors then study the generalization of GNN based on kernel-graph alignment.

**Strengths:**

(1) The connection between the learning dynamics of GNN and label propagation via residual propagation is novel and insightful.

(2) The theoretical analysis is deep and elegant.

**Weaknesses:**

(1) The assumption of infinitely wide network is not realistic. It is better to analyze the evolution of the kernel.

(2) The restriction to two-layer GNN or last-layer feature propagation is not realistic either.

**Questions:**

Is it possible to go beyond neural tangent kernel and two-layer GNN? What theoretical tools are needed? I assume there are methods developed for MLP.

---

> ### Author Response · Authors · 2023-11-16
> **Response to Reviewer ZpWJ (Part 1/1)**
>
> Thanks for the constructive feedback. We address each point as follows.
>
> **`Weakness.1`: "The assumption of infinitely wide network is not realistic. It is better to analyze the evolution of the kernel."**
>
> Though idealized, the infinite width limit or NTK regime is widely used for shedding insight into problems that are otherwise highly prohibitive to analyze. Previous studies for the NTK regime have led to a wealth of advances in numerous problems such as dataset distillation, matrix completion, generalization prediction, architectural search (see [1] and references therein). In our particular scenario, the insights drawn from the infinite width regime make predictions regarding behavior in more practical finite-width models. Importantly, these predictions can be largely verified based on extensive empirical verification spanning 17 datasets.  So indeed, although as the reviewer suggests infinitely wide models may be idealized, they still have practical value as an analysis tool, and our theoretical results still asymptotically hold for larger models in the finite width case.
>
> **(Evolution of NTK)** As for evolution of the kernel in the finite width case, a general theoretical characterization is still largely infeasible [2]. Nevertheless, we have provided empirical analysis of the evolution of the NTK during training. As elaborated in Section 5.2, our observations on the kernel evolution include: 1) the NTK of GNNs tend to align with the graph; 2) the NTK also tends to align with the optimal kernel as it evolves during the training process, and heterophily hinders this process since alignment to both graph and the optimal target become contradictory.
>
> [1] Neural Tangent Kernel: A Survey, 2022
>
> [2] Spectral Evolution and Invariance in Linear-width Neural Networks (ICML 2023 Workshop)
>
>
>
>
> **`Weakness.2`: "The restriction to two-layer GNN or last-layer feature propagation is not realistic either."**
>
> In Section 4.1, we have derived the explicit formula for the node-level GNTK, which is compatible with any GNN defined in Section 2. Such a formula demonstrates how the graph adjacency is integrated into the kernel similarity measure. The purpose of Section 4.2 is then to provide concrete examples for strengthening and illustrating these results, and thus architectural restrictions are necessary.
>
> Since there is typically no explicit, analytically-tractable expression of the NTK for very deep NNs (only a formula for recurrently computing it exists), we consider two-layer GNNs in many previous works on MLPs [3]. In addition to this however, we introduce a notable exception where there indeed exists an explicit NTK expression for deep GNNs with learnable embedding inputs and feature propagation at the last layer, which is why we consider this type of architecture. Moreover, feature propagation at the last layer is not uncommon; many popular GNNs follow this architecture such as APPNP [4] and GPR-GNN [5]. We hope this helps to clarify, but we are happy to provide further details if needed.
>
> [3] Fine-Grained Analysis of Optimization and Generalization for Overparameterized Two-Layer Neural Networks (ICML 2019)
>
> [4] Predict then propagate: Graph neural networks meet personalized pagerank (ICLR 2019)
>
> [5] Adaptive Universal Generalized Pagerank Graph Neural Network (ICLR 2021)
>
>
>
>
> **`Question`: "Is it possible to go beyond neural tangent kernel and two-layer GNN? What theoretical tools are needed? I assume there are methods developed for MLP."**
>
> **(Beyond NTK Regime)** Going beyond the NTK regime may be possible in principle; however, the analysis either still requires certain degrees of overparameterization, leading to imprecise results based on bounds that are heavily dependent on the width (e.g. [5]), or uses other approximations such as second-order Taylor expansion of neural networks (e.g. [6]). Unfortunately, exact mathematical characterization of the optimization for finite width MLPs, even for those shallow ones, is still an open question in general neural network theory. In fact, there are quite a few recent papers just on analyzing single-neuron networks (e.g. [7,8]), indicating the difficulty of analyzing not-so-wide networks.
>
> **(Beyond Two-Layer)** Extending to deeper layers is somewhat more feasible. We have provided a formula for computing the NTK of arbitrarily deep GNNs in Section 4.1. However, the expression of the NTK for deeper neural networks is quite complex and not conducive to further analysis at this point. Still this is an interesting direction we hope to explore in future work if possible.
>
> [6] Learning Over-Parameterized Two-Layer ReLU Neural Networks beyond NTK (COLT 2020)
>
> [7] Beyond Linearization: On Quadratic and Higher-Order Approximation of Wide Neural Networks (ICLR 2020)
>
> [8] Agnostic learning of a single neuron with gradient descent (NeurIPS 2021)
>
> [9] Learning a Neuron by a Shallow ReLU Network: Dynamics and Implicit Bias for Correlated Inputs (2023)

---

### Official Review · Reviewer_mEP4 · 2023-11-04

**Soundness:** 3 good
**Presentation:** 3 good
**Contribution:** 3 good
**Rating:** 6
**Confidence:** 2

**Summary:**

This paper studies the training dynamics and generalization of graph neural networks (GNNs). The authors theoretically derive the evolution of the residuals of GNNs on training and testing data in several settings and based on this, they explain the generalization ability of GNNs. Some numerical verification is also reported.

**Strengths:**

1. The analysis of training dynamics and generalization is an extremely important topic in the research of GNNs. This paper has a good scope and is clearly written to connect ideas from different fields.
2. Although I think there are some limitations, the derived theoretical results are technically solid and are pleasing to read. The authors use tools from label propagation and graph neural tangent kernel to characterize the training dynamics of GNNs and they derive explicitly dynamics in several cases.
3. The authors give some reasonable explanation of the GNN generalization by connecting GNN training dynamics and optimal kernel.

**Weaknesses:**

1. In Section 4.2, the authors only consider two very special $\bar{\mathcal{X}}$, which makes the theory somehow limited.
2. The training dynamics of GNNs should be highly nonlinear. More explicitly, in equation (12), the GNTK $\Theta_t^{(l)}$ depends on $W_t$. However, the derived dynamics in Theorem 5 and Theorem are linear. The authors need to explain how they remove the nonlinearity and why it makes sense.

**Questions:**

1. This question is related to the second point in the "Weakness". According to your derivations in Section B.4, I think you remove the nonlinearity or the dependence of the kernel on the parameters $W$ by taking the expectation for $W$. Please correct me if I misunderstood something. I am confused as to why you take the expectation -- in my opinion, training dynamics is the evaluation of residuals or parameters for any given/fixed initialization. If you want to take expectation over $W$, I think the equation (14) and (15) should be stated as something like expected residual. Please explain what is happening and why it is reasonable.
2. In Theorem 5, do you have the training dynamics for optimizing $W^{(2)}$? In Theorem 6, is the training dynamics for optimizing all parameters in the GNN, or it is just optimizing parameters in a single layer (as in Theorem 5)?

---

> ### Author Response · Authors · 2023-11-16
> **Response to Reviewer mEP4 (Part 1/2)**
>
> Thanks for the constructive feedback. We address each point as follows.
>
> **`Weakness.1`: "In Section 4.2, the authors only consider two very special $\bar{\mathcal{X}}$, which makes the theory somehow limited."**
>
> In Section 4.1, we have given the general form of GNN training dynamics where the NTK can be derived based on our explicit formula. This part of the analysis is compatible with an arbitrary $\bar{\mathcal{X}}$ and demonstrates how the graph adjacency is naturally integrated into the kernel similarity measure. Section 4.2 complements 4.1 by providing concrete examples, for which purpose constraints on inputs are necessary. In comparison, previous work on GNN expressiveness often assume the graph is endowed with node-coloring [1], while other work assumes node features and graph structure are generated by a probabilistic model such as CSBM [2].
>
> Moreover, we should also mention that the choices for $\bar{\mathcal{X}}$ are not arbitrary or contrived, but closely related to practice: $\bar{\mathcal{X}} = \mathbf I$ represents learnable node embeddings that are ubiquitous in GNN applications such as recommender systems and knowledge graphs, while $\bar{\mathcal{X}}$ from a graph spectral decomposition unifies many network embedding methods such as DeepWalk and Node2Vec, which themselves are being actively researched.
>
> [1] Theory of Graph Neural Networks: Representation and Learning (2022)
>
> [2] Understanding Non-linearity in Graph Neural Networks from the Perspective of Bayesian Inference (NeurIPS 2022)
>
> **`Weakness.2`: "The training dynamics of GNNs should be highly nonlinear. More explicitly, $\mathbf \Theta_t$ depends on $\mathbf W$. However, the derived dynamics in theorems are linear."**
>
> This is an important question. The answer is related to NTK theory, which we leverage to translate an intractable problem into a tractable one. While the training dynamics of NN (neural network) models are generally nonlinear, recent analysis has shown that they asymptotically behave as linear models as width increases. For example, Theorem 2 in the seminal NTK work [3] states that NTK $\mathbf \Theta_t$ at arbitrary time $t$ converges (w.r.t. width) to a constant kernel function at initialization $\mathbf \Theta_0$. This result holds universally for other architectures [4]. Proofs of such results are usually mathematically intense, and counter-intuitive as the reviewer pointed out. Informally, it can be understood as follows: if the width goes to infinity, the movement of parameters caused by optimization is such that the function $f(x)$ itself changes, but perhaps counter-intuitively, the Jacobian and NTK do not change appreciably. This explains the dissociation of $\mathbf \Theta_t$ and $\mathbf W$.
>
> For the linearization, it essentially stems from the first order Taylor expansion of a model around the initialization [5], i.e. $f(x) \approx f_0(x) + \nabla_{\mathbf{W}} f_0(x)^\top (\mathbf W - \mathbf W_0)$, which is indeed a linear model w.r.t. $\mathbf W$ that approximates the original $f(x)$. Such an approximation becomes precise for infinite width NNs, since the Jacobian stays constant and the Hessian can be proven to approach zero [6]. And because both $\nabla_{\mathbf{W}} f_0(x)^\top$ and $\mathbf W$ are in $\mathbf R^{\infty}$, NN optimization becomes equivalent to kernel regression [3]. That being said, it is important to note that the nonlinearity in the model is preserved in the derivative, and thus the benefits that GNNs can obtain from nonlinear activations are also preserved in our analysis.
>
> [3] Neural Tangent Kernel: Convergence and Generalization in Neural Networks (NeurIPS 2018)
>
> [4] Tensor programs iib: Architectural universality of neural tangent kernel training dynamics (ICML 2021)
>
> [5] Wide Neural Networks of Any Depth Evolve as Linear Models Under Gradient Descent (NeurIPS 2019)
>
> [6] On the linearity of large non-linear models: when and why the tangent kernel is constant (NeurIPS 2020)

---

> ### Author Response · Authors · 2023-11-16
> **Response to Reviewer mEP4 (Part 2/2)**
>
> **`Question.1`: "I think you remove the nonlinearity or the dependence of the kernel on the parameters $\mathbf W$, by taking the expectation for $\mathbf W$"; "I am confused as to why you take the expectation -- in my opinion, training dynamics is the evaluation of residuals or parameters for any given/fixed initialization."**
>
> The expectation stems from computing the NTK at its initialization where there are infinitely many parameters sampled from a Gaussian, the weighted sum of which induces expectation. To be more specific, as mentioned above, NTK $\mathbf \Theta_t$ at arbitrary time $t$ converges (w.r.t. model width) to a constant kernel function at its initialization $\mathbf \Theta_0$, and thus we need only to compute the $\mathbf \Theta_0$ where $\mathbf W^{(1)}$ is Gaussian distributed (as stipulated by the standard NTK parameterization stated in theorems). In the computation, expectation emerges from the fact that $\mathbf W^{(1)} = [w_1, w_2, \cdots, w_{\infty}] \in \mathbb R^{d\times \infty}$ is a concatenation of infinitely many vectors from $\mathcal N(0,\mathbf I_d)$, with a scaling factor $1/m$ (and $m\rightarrow \infty$) in the NTK for averaging. Similar results can be found for two-layer MLPs [7].
>
> [7] Fine-Grained Analysis of Optimization and Generalization for Overparameterized Two-Layer Neural Networks (ICML 2019)
>
>
>
> **`Question.2`: "In Theorem 5, do you have the training dynamics for optimizing $\mathbf W^{(2)}$ ? In Theorem 6, is the training dynamics for optimizing all parameters in the GNN?"**
>
> Yes, Theorem 5 can be extended straight-forwardly to incorporate $\mathbf W^{(2)}$, i.e. $\nabla_{\mathbf{W}} \mathcal{F}^{\top} \nabla_{\mathbf{W}} \mathcal{F} = \nabla_{\mathbf{W^{(1)}}} \mathcal{F}^{\top} \nabla_{\mathbf{W^{(1)}}} \mathcal{F} + \nabla_{\mathbf{W^{(2)}}} \mathcal{F}^{\top} \nabla_{\mathbf{W^{(2)}}} \mathcal{F}$. The latter one is relatively easy to derive as $\mathbf{W^{(2)}}$ is applied to the last layer. Consequently, the propagation matrix in Theorem 5 should be revised to $\mathbf{A}\left(\mathbf{A}^2 \odot \mathbf{S} + \mathbf A^\prime \right) \mathbf{A}$, where  $\mathbf A^\prime_{ij} = \mathbb{E}_{{w} \sim\mathcal{N}(0,\mathbf I)}[\sigma(w^\top \mathbf A_i) \sigma(w^\top \mathbf A_j)]$  measures similarity of neighborhood patterns for two nodes. We did not originally consider $\mathbf{W^{(2)}}$ because it is often omitted by convention. All other analyses (including Theorem 6) consider optimizing all parameters.

---

### Official Review · Reviewer_mA7d · 2023-11-05

**Soundness:** 3 good
**Presentation:** 3 good
**Contribution:** 2 fair
**Rating:** 5
**Confidence:** 3

**Summary:**

This paper investigates the function-space learning dynamics of graph neural networks (GNNs) during gradient descent. The key contributions include:

-   Identifying the similarity between GNN learning dynamics and cross-instance label propagation, facilitated by the neural tangent kernel (NTK).
-   Theoretical insights into why GNNs demonstrate strong generalization on graphs with high homophily, connected to NTK’s natural alignment with graph structure.
-   Development of a Residual Propagation (RP) algorithm inspired by these dynamics, showcasing notable performance improvements over standard GNNs.
-   Examination of GNN limitations on heterophilic graphs, including empirical validation on both synthetic and real-world datasets, revealing misalignments between NTK and the graph structure.

**Strengths:**

**Originality**: While connecting GNN dynamics to propagation schemes is novel, the paper lacks some innovation in terms of proposing new techniques beyond the basic RP algorithm. Theoretical insights relate to existing works on kernel alignment and generalization.

**Quality**: The theoretical claims rely heavily on assumptions of overparameterization and alignment of NTK with adjacency matrix, which may not perfectly hold in practice. More analysis is needed for finite width GNNs. Empirical evaluation is quite limited.

**Clarity**: The key ideas are reasonably clear, the significance of results is not fully crystallized.

**Significance**: The insights on generalization are incremental on existing theory on kernel alignment. Practical impact is unclear given the simplicity of RP and lack of evaluation on large benchmarks. The limitations of GNNs on heterophily are already well-known.

**Weaknesses:**

**Strong assumptions**: The study's theoretical framework is built on robust assumptions regarding infinite width and NTK alignment that may not hold across all scenarios. Expanding the analysis to cover finite-width GNNs could substantiate the findings.

**Limited evaluation**: The empirical validation is limited in scope, focusing on smaller datasets and simpler models. Extensive testing involving state-of-the-art GNNs and more diverse benchmarks would be instrumental in corroborating the theoretical claims.

**Significance in theory**:  The theoretical contributions, while valuable, seem to offer only a modest advancement beyond existing studies on kernel alignment. Clarifying the distinctions from previous work would help to highlight the unique contributions of this study.

**Questions:**

-   How does the theoretical analysis diverge from previous studies on kernel alignment? Clarification of the novel insights would be appreciated.
-   Given the recognized challenges of GNNs in dealing with heterophily, are there any strategies or recommendations proposed by the authors to tackle this issue beyond the current analysis?

---

> ### Author Response · Authors · 2023-11-16
> **Response to Reviewer mA7d (Part 1/3)**
>
> Thanks for the constructive feedback. As the reviewer concerns are spread across strengths (S), weaknesses (W), and questions (Q), we will organize and address them grouped by topic.
>
> **`Q.1`: "How does the theoretical analysis diverge from previous studies on kernel alignment?"; `S.Significance`: "The insights on generalization are incremental on existing theory on kernel alignment."; `W.Significance in theory`: "The theoretical contributions, while valuable, seem to offer only a modest advancement beyond existing studies on kernel alignment."**
>
> **(Pior Work on Kernel Alignment)** Kernel-target alignment [1] is a well-established concept, with application to kernel methods, e.g., as an objective for learning kernels [2]. However, the connection with generalization is mostly restricted to classic models such as Parzen window estimators [3]. Overall, this research field has been active well before the era of deep learning, and subsequently, established connections to deep learning models are loose, e.g., prior work applying kernel alignment to deep model generalization does not exist to the best of our knowledge. Importantly though, we do not aim to make general advancements in this classic field itself.
>
> **(Ours)** Instead, we repurpose the generic notion of kernal alignment within a completely new use-case, namely, quantifying and demystifying the generalization performance of neural network architectures, and GNNs in particular. To this end, we introduce the novel construct of kernel-graph alignment, which is quite distinct from classical kernel alignment.  We then leverage this construct to generate a series of significant insights regarding the evolution of GNN functions during optimizaiton, informing when and why the resulting learned functions can generalize.
>
> Beyond this, we are unaware of existing work that elucidates neural network architectural designs (GNNs or otherwise) through the lens of kernel alignment as we have done.  That being said, we are happy to contextualize with respect to any additional references the reviewer might refer us to.
>
> [1] On kernel-target alignment (NeurIPS 2001)
>
> [2] Learning the kernel matrix with semidefinite programming (JMLR 2004)
>
> [3] Remarks on Some Nonparametric Estimates of a Density Function (1956)
>
>
> **`Q.2`: "Are there any strategies or recommendations proposed by the authors to tackle this issue [heterophily] beyond the current analysis?"; `S.Significance`: "The limitations of GNNs on heterophily are already well-known."**
>
> **(Recommendation for Heterophily)** While recommending specific GNN architectures for tackling heterophily is an active research field mostly beyond our scope, we can nonetheless recommend more high-level design principles motivated by our analysis:
> 1. Increasing $A(\mathbf A, \mathbf \Theta^*)$ by designing propagation matrices that deviate from the original adjacency matrix, such that the resulting GNN NTK could better align with the target, can lead to better generalization performance.  Moreover, this perspective helps to substantiate certain existing heuristics, such as assigning negative weights to edges.
> 2. Decreasing $A(\mathbf \Theta_t, \mathbf \Theta^*)$ by designing non-standard GNN architectures, i.e., that deviate from our defintion in Section 2, can reduce the degradation of NTK-target alignment caused by heterophily; in doing so, the negative effects of heterophilic graphs on generalization can also be mitigated. Incidently, this also helps to quantify regimes where non-GNN architectures such as MLPs can sometimes perform better.
>
>
> We should also mention that even the extremely simple RP algorithm that naturally emerges from our analysis works well in practice handling heterophily.  See Section E.3 of the supplementary for empirical examples.
>
> **(Significance of Connection with Heterophily)** Although it is already well-known from empirical observations that GNN performance may degrade at times on heterophilic graphs, our analysis is still quite relevant in providing principled, complementary explanations for why and how this happens.

---

> > ### Author Response · Authors · 2023-11-16
> > **Response to Reviewer mA7d (Part 2/3)**
> >
> > **`S.Originality`: "The paper lacks some innovation in terms of proposing new techniques beyond the basic RP algorithm."**
> >
> > **(New Techniques for RP)** We kindly note that the RP algorithm itself is an original learning paradigm first proposed in this work, which serves to demonstrate a surprising phenomenon: directly replacing the NTK with the graph adjacency matrix can lead to an algorithm with GNN-level performance. This phenomenon strongly supports our theoretical result that graph-kernel alignment is critical for explaining GNN generalization performance. Proposing new techiniques does not contribute to such a purpose, and thus overall is not our focus.
> >
> > Nevertheless, we do agree that combining with other orthogonal techniques, which are abundant in the graph learning literature, could be promising. Thus we have provided: **1)** a fairly general form of RP in equation 9 allowing endless new designs that practitioners could potentially pursue; **2)** an instantiaton of equation 9 by combining it with kernel functions in Appendix E.
> >
> >
> >
> > **`S.Quality`: "The theoretical claims rely heavily on assumptions of overparameterization and alignment of NTK with adjacency matrix, which may not perfectly hold in practice. More analysis is needed for finite width GNNs. Empirical evaluation is quite limited."; `W.Strong assumptions`: Similar comments as in S.Quality.**
> >
> > **(Overparameterization Assumption)** This is a good point and worth clarifying further here. While insights drawn from formally analyzing finite-width cases would ideally be nice to have, unfortunately in general settings this is prohititively difficult and remains a major challenge in neural network theory. As such, it is a common practice to simplify to overparameterized settings to make the problem analytically tractable (see [4] and numerous references therein). This mirrors theoretical results more broadly, where there is typically a gap between theory and practice. But theory nonetheless maintains relevance if it makes predictions that can be confirmed empirically despite deviations from the original assumptions. In this regard, our insights from the overparameterized setting make testable predictions w.r.t. intractable finite width models that we verify empirically, e.g., Tables 1,3,4,5 and Figures 2,3,4 spanning 17 datasets (admittedly, some of these were deferred to the supplementary for space considerations).
> >
> > **(Alignment "Assumption")** Unlike the overparameterized simplification mentioned above, alignment of the NTK with the graph (adjacency matrix) is **not** actually a general assumption of our analysis.  Rather, NTK-graph alignment is a novel concept that we have introduced as an analysis lens through which we can provide interpretable answers to when and why GNNs generalize.  And it is only for equation 16 in particular that we assume perfect alignment for the purpose of deriving a succinct generalization bound; elsewhere no such perfect alignment is assumed.
> >
> > [4] Neural Tangent Kernel: A Survey (2022)
> >
> >
> >
> >
> > **`S.Clarity`: "The significance of results is not fully crystallized."**
> >
> > **(Our Significance)** The significance of our results manifests as providing interpretable answers to "how GNNs learn" (optimizaton) and "why they succeed" (generalization). These concepts are fundamental to any learning system, but yet somehow have been largely overlooked for GNNs (especially optimization). Moreover, these insights translate into a practically-useful white-box RP algorithm capable of more efficiently outperforming standard GNNs on popular benchmarks.
> >
> >
> > **`S.Significance`: "Practical impact is unclear given the simplicity of RP and lack of evaluation on large benchmarks."**
> >
> > **(Practical Impact of RP)** We would argue that RP is practically impactful precisely because of its simplicity and interpretability. According to results on multiple OGBN datasets in Table 1 (some of the larger public benchmarks available, although scalability is not our focus herein), this simple RP approach generally outperforms widely-used GNNs, with up to $10^4 \times$ speed up, and $10\times$ less memory. Strong performance is also shown on another 12 datasets in Tables 4 and 5 of the supplementary. Such efficiency is generally unparalleled among methods with GNN-level performance. Moreover, because RP is simple and transparent, we can characterize why and when it works, with multiple entry points for flexibly combining with orthogonal techniques to further improve performance.

---

> > > ### Author Response · Authors · 2023-11-16
> > > **Response to Reviewer mA7d (Part 3/3)**
> > >
> > > **`W.Limited evaluation`: "The empirical validation is limited in scope, focusing on smaller datasets and simpler models. Extensive testing involving state-of-the-art GNNs and more diverse benchmarks would be instrumental in corroborating the theoretical claims."**
> > >
> > > **(Scale and Diversity of Datasets)** We remark that our empirical verification spans 17 datasets (Arxiv, Proteins, Products, Cora, Citeseer, Pubmed, Computers, Photo, CS, Physics, Roman-empire, Amazon-ratings, Minesweeper, Tolokers, Questions, Texas, Synthetic), including homophilic, heterophilic, and up-to-date (proposed in 2023) examples. Even so, we concede that some of these were deferred to the supplementary to save space and could easily be missed.  We could reorganize to include more in the main paper if the reviewer suggests and space allows.
> > >
> > > Additionally, with respect to size, the OGBN datasets in Table 1 are recognized to be relatively large-scale benchmarks, with up to 2,449,029 nodes and 61,859,140 edges.  Although there do exist a few larger benchmarks, dealing with these requires considerable engineering effort and compute, and scalability is not our primary focus. Overall then, we would argue that for an analysis-oriented paper in particular, our experimental coverage is considerable, especially so if we account for the supplementary.
> > >
> > > **(SOTA Models)** As for baseline models for comparison, our experiments are specifically designed to complement and corroborate analytical results, not for challenging an ever-changing landscape of SOTA models.  That being said, in Appendix E.3, we have evaluated RP against an array of recent GNNs explicitly designed to handle heterophily, and it still performs competitively despite its far greater simplicity.

---

### Meta-Review · Area_Chair_qCYf · 2023-12-12

**Metareview:**

Summary: The article investigates the training dynamics of GNNs in function space and generalization.

Strengths: Referees found the paper had a good scope, included theoretical results and explanations of generalization in GNNs. One of the referees found the connection between learning dynamics of GNNs and label propagation novel and insightful.

Weaknesses: On the critical side, a referee found the work relies heavily on certain assumptions (alignment of NTK and adjacency), that the results could be regarded as incremental, and that the practical impact was not sufficiently explained. Another referee had concerns about limitations of the theory and the formulation of the dynamics. Some of these limitations were echoed by another referee, who found assumptions and restrictions unrealistic.

At the end of the discussion period this article had mixed ratings, with two referees regarding it as borderline and another as good. Upon inspecting the article, I find that although it presents interesting and promising new ideas, the theoretical analysis is indeed restrictive and the proposed label-propagation view of GNN dynamics and NTK adjacency alignment could be strengthened by more extensive investigation. Given the high acceptance bar I must reject the article at this time.

**Justification For Why Not Higher Score:**

The theoretical analysis builds on unrealistic assumptions. More extensive empirical evaluations could still strengthen the proposed perspectives.

**Justification For Why Not Lower Score:**

NA

---

### Decision · Program_Chairs · 2024-01-16

Reject